# Introducing Sample Robustness

## Abstract

Choosing the right data and model for a pre-defined task is one of the critical competencies in machine learning. Investigating what features of a dataset and its underlying distribution a model decodes may enlighten the mysterious "black box" and guide us to a deeper and more profound understanding of the ongoing processes. Furthermore, it will help to improve the quality of models which directly depend on data or learn from it through training. In this work, we introduce the dataset-dependent concept of sample robustness, which is based on a point-wise Lipschitz constant of the label map. For a particular sample, it measures how small of a perturbation is required to cause a label-change relative to the magnitude of the label map. We introduce theory to motivate the concept and to analyse the effects of having similar robustness distributions for the training- and test data. Afterwards, we conduct various experiments using different datasets and (non-)deterministic models. In some cases, we can boost performance by choosing specifically tailored training(sub)sets and hyperparameters depending on the robustness distribution of the test(sub)sets.

## 1 Introduction:

In the age of automated machine learning, we shift our focus evermore towards regarding meta-hyperparameters such as model-type or training- and validation budget as variables of a loss function in the most abstract sense. For training sets, however, the mere number of samples often determines how well suited it is perceived for a particular task. The motivation of this work is to introduce a concept allowing for the use of datasets as variables of such a generalised loss function as well.

Imagine, for example, patients who share almost identical medical records, but react differently to some prescribed treatment. They may pose a challenge to machine learning models similar to what is known as natural adversarial examples in vision tasks, see Hendrycks et al. (2019). The relationship between medical features and appropriate personal treatments may be sensitive towards small input variations, i.e. not robust towards perturbations. In this work, we are to the best of our knowledge the first to introduce and analyse a model-agnostic measure for the robustness of data. We show that knowledge about the robustness distribution of a specific test(sub)set can allow for choosing a more appropriate training(sub)set in terms of performance optimisation. Finally, we discover that the optimal choice of hyperparameters may also depend on the robustness distributions of both training- and test data.

Let us motivate the concept of sample robustness first on a high level. When collecting and processing a dataset for a pre-defined task, we identify certain features and expressive samples such that a model may be able to abstract and generalise from these finite points to the whole space of possible inputs. Assume we have a certain rectangle-shaped data-distribution in a circle-shaped feature space and a dataset labelled according to two distinct ground truth maps $y^* \in \{\times, *\}$ and $z^* \in \{\circ, \bullet\}$ (comp. **Figure 1**). Here, one can imagine classifying images of horses and cats (assuming ground truth $y^*$) and classifying images of animal-human pairs (assuming ground truth $z^*$). Evidently, the distance between differently labelled samples depends on the ground truth map labelling them.

For every sample in a dataset, the intrinsic information of closeness to a differently labelled sample can be considered a feature itself. For regression tasks and label distributions which are not necessarily categorical one may also include the distance of the corresponding labels as additional information. By taking the quotient of these two and maximising it over the dataset, i.e. calculating a point-wise Lipschitz constant of the label map, one can measure how sensitive a sample is to label-

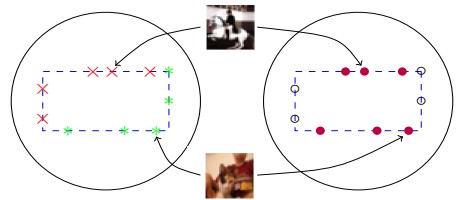

Figure 1: Images from the CIFAR-10 dataset labelled "horse" and "cat", respectively.

changing perturbations. Precisely this is the non-formal basis of the concept we propose, namely sample robustness. Our work aims to analyse the connection between both model performance and aligning the robustness distributions of training- and test sets.

## 1.1 OUTLINE

After citing and discussing related work concerned with decision boundaries, model robustness and Lipschitz calculus in section 2, we introduce the mathematical framework and the measure of sample robustness in section 3. We also motivate the concept theoretically and show some natural relations to K-Nearest Neighbour models. Section 4 is completely devoted to the evaluation using different datasets (CIFAR-10, Online News Popularity Dataset) and models (Convolutional Neural Networks, K-Nearest Neighbour, Random Forest). Section 5 finally concludes the findings and discusses other research approaches. Letters **A** - **F** refer to sections in the appendix.

## 2 RELATED WORK:

Analysing the data-distribution before training yields a way to investigate (and boost) model performance from an earlier stage as is done by unsupervised pre-training (Erhan et al., 2010)). Many algorithms stemming from the unsupervised setting (Chigirev & Bialek, 2003; Cayton, 2005) are devoted to extract information about the so-called data-manifold (Fefferman et al., 2013; Bernstein & Kuleshov, 2014). Decoding the latent features which determine the data-distribution (Bengio et al., 2013) provides valuable insight and helps to understand the decision boundaries which a model learns throughout the training phase. Furthermore, understanding the data-manifold may provide a view into the "black box" transforming inputs to outputs (Fawzi et al., 2016; Liu et al., 2017; Biggio et al., 2013). In this work, we want to use the intrinsic information of distance between samples in feature space and relate it to the distance of the corresponding labels to introduce a new dataset-dependent feature.

The robustness of a sample can be regarded as its susceptibility to label-changing perturbations. Here, one is immediately reminded of adversarial examples (Szegedy et al., 2013) in the context of model robustness. The difference to our proposed concept, however, is that we only use the pre-defined labels instead of model predictions as additional input. The term robustness itself is one of the most prominent throughout the recent literature in many different contexts, from robust attacks to robust models/defences (Evtimov et al., 2017; Beggel et al., 2019; Madry et al., 2017; Weng et al., 2018; Tsuzuku et al., 2018). State-of-the-art machine learning models are susceptible to noise, especially so when crafted purposefully (Fawzi et al., 2016). It leaves these powerful machines vulnerable to attacks either at training- (Zhu et al., 2019) or at test stage (Biggio et al., 2013), independent of the architecture used (Papernot et al., 2016b). We follow the idea that models are extensions of the label map from the (metric) subspace defined by a dataset to the whole feature space. Hence, they will inherit critical properties from the data. In this work we analyse the robustness distribution of datasets and the thereon dependent performances of models, but plan on investigating the connection to model-robustness in the future.

Lipschitz calculus yields a mathematically well-understood approach to describe and measure model robustness as in Weng et al. (2018) or Tsuzuku et al. (2018). Framing machine learning theory in terms of metric spaces (and also building robustness concepts thereon) has been done before in Wang et al. (2016), however, not explicitly connecting it to Lipschitz calculus. In this work, we

build the concept of sample robustness based on a point-wise Lipschitz constant of the label map for metric- and Banach spaces such that it applies to a wide range of different feature- and label spaces including an ample variety of metrics.

Finally, investigating the dependence of model-hyperparameters on data has been done previously (Nakkiran et al., 2020). The authors also showed that more data could sometimes decrease model performance. In this work, we will see similar results regarding these two aspects with the difference that we could identify such data even before training using the proposed measure of sample robustness.

## 3 SAMPLE ROBUSTNESS

Now we will introduce the primary concept of this work, namely sample robustness. It measures how sensitive samples are to label-changing perturbations relative to the magnitude of the label map.

### 3.1 DEFINITION: (FRAMEWORK)

Let $\mathcal{FS}$ be a feature space, i.e. a metric space with metric $\mathrm{d} := \mathrm{d}_{\mathcal{FS}}$, let $\mathcal{TS}$ be a target- or label space, i.e. a real Banach space with norm $\|\cdot\| := \|\cdot\|_{\mathcal{TS}}$, let $\mathbf{x}, \mathbf{t} \subseteq \mathcal{FS}$ be finite (data)sets with $\#\mathbf{x}, \#\mathbf{t} \geq 2$ and let $y : \mathbf{x} \cup \mathbf{t} \to \mathcal{TS}$ be a map of labels with $\#y(\mathbf{x}) = \#y(\mathbf{t}) \geq 2$.

### 3.2 DEFINITION: (REACH)

Let $x \in \mathcal{FS}$ be a sample with label $y(x)$, $\|y(x)\| \leq \|y\|_\infty$, and $\Gamma_\mathbf{x}(x) := \{\tilde{x} \in \mathbf{x} \mid y(x) \neq y(\tilde{x})\}$. The **reach** of $x$ is defined as the distance of $x$ to $\Gamma_\mathbf{x}(x)$:

$$\mathbf{r}_\mathbf{x}(x) := \mathrm{dist}\big(x, \Gamma_\mathbf{x}(x)\big) = \min_{\tilde{x} \in \Gamma_\mathbf{x}(x)} \mathrm{d}(x, \tilde{x})$$

In other words: $\Gamma_\mathbf{x}(x)$ is the set of samples $\tilde{x}$ in the dataset $\mathbf{x}$ that are labelled differently from $x$. One may notice that $\tilde{x} \in \Gamma_\mathbf{x}(x) \Leftrightarrow x \in \Gamma_\mathbf{x}(\tilde{x})$. The reach of $x \in \mathbf{x}$ is exactly the minimal distance of the point representing the image of a "cat" to a differently labelled (coloured) point in **Figure. 1**. If $x$ is "close" to $\tilde{x}$, one would expect their labels $y(x)$ and $y(\tilde{x})$ to be "close" as well. Taking the quotient then gives a measure of this error (comp. 3.3) and normalising the latter with respect to the magnitude of $y$ (using $\|y\|_\infty := \max_{x \in \mathbf{x}} \|y(x)\|$) yields the main concept of sample robustness in 3.4.

### 3.3 DEFINITION: (POINT-WISE LIPSCHITZ CONSTANT)

For $x \in \mathcal{FS}$ with label $y(x)$, $\|y(x)\| \leq \|y\|_\infty$, and $\Gamma_\mathbf{x}(x)$ as in **3.2** one defines a point-wise Lipschitz constant of $y$ as:

$$\mathcal{Q}_\mathbf{x}(x) := \max_{\tilde{x} \in \Gamma_\mathbf{x}(x)} \frac{\|y(x) - y(\tilde{x})\|}{\mathrm{d}(x, \tilde{x})}$$

### 3.4 DEFINITION AND PROPOSITION: (SAMPLE ROBUSTNESS)

Let $x \in \mathcal{FS}$ with label $y(x)$, $\|y(x)\| \leq \|y\|_\infty$, and $\Gamma_\mathbf{x}(x)$ as in **3.2**. The **robustness of the sample** $x$ **in** $\mathbf{x}$ with respect to $\mathrm{d}$ is defined as:

$$\mathcal{R}_\mathbf{x}(x) := \frac{\|y\|_\infty}{\mathcal{Q}_\mathbf{x}(x) + \|y\|_\infty} \in (0, 1)$$

It is independent of rescaling the label map $y$ (comp. **A.2**). Coming back to the example in the introduction, we can see now that the almost identical medical records of patients reacting differently to the same treatment are considered as less robust samples in the above sense.

### 3.5 Theoretic Motivation and Background

Assume we have datasets $\mathbf{x}$ and $\mathbf{t}$ where both are labelled using the same label map $y$ with $\max_{x \in \mathbf{x}} \|y(x)\| = \max_{t \in \mathbf{t}} \|y(t)\|$. For any $z \in \mathbf{x} \cup \mathbf{t}$ it holds that:

$$\mathcal{Q}_{\mathbf{x} \cup \mathbf{t}}(z) = \max\{\mathcal{Q}_{\mathbf{x}}(z), \mathcal{Q}_{\mathbf{t}}(z)\} \Leftrightarrow \mathcal{R}_{\mathbf{x} \cup \mathbf{t}}(z) = \min\{\mathcal{R}_{\mathbf{x}}(z), \mathcal{R}_{\mathbf{t}}(z)\}$$

In other words: the closer $\mathcal{R}_{\mathbf{x}}(z)$ is to $\mathcal{R}_{\mathbf{t}}(z)$, the closer both values are to the robustness of $z$ in the union $\mathbf{x} \cup \mathbf{t}$. It follows:

$$\mathcal{R}_{\mathbf{x}}(z) \approx \mathcal{R}_{\mathbf{t}}(z) \Rightarrow \mathcal{R}_{\mathbf{x}}(z) \approx \mathcal{R}_{\mathbf{x} \cup \mathbf{t}}(z) \approx \mathcal{R}_{\mathbf{t}}(z),$$

where at least one side is an equality. For convenience, we write $\mathbf{x} \sim_{\mathcal{R}} \mathbf{t} :\Leftrightarrow \mathcal{R}_{\mathbf{x}}(z) \approx \mathcal{R}_{\mathbf{t}}(z) \;\; \forall\, z \in \mathbf{x} \cup \mathbf{t}$[1]. Assume now that $F$ is an extension of the label map $y$ from $\mathbf{x}$ to $\mathbf{x} \cup \mathbf{t}$[1]. For given $\mathbf{t}$ one can downsize the set $\mathbf{x}$ to $\tilde{\mathbf{x}} \subset \mathbf{x}$ in order to align both robustness distributions; however, there will likely be a trade-off between this alignment and the distance of $F$ to $y$ as maps on $\mathbf{t}$, because there are less points to extend from (therefore allowing for a higher variance).

For such an extension $F$ it holds that $F_{|\mathbf{x}} \equiv y_{|\mathbf{x}}$, thus:

$$\mathcal{Q}_{\mathbf{x}}(x) = \max_{\tilde{x} \in \Gamma_{\mathbf{x}}(x)} \frac{\|y(x) - y(\tilde{x})\|}{\mathrm{d}(x, \tilde{x})} = \max_{\tilde{x} \in \Gamma_{\mathbf{x}}(x)} \frac{\|F(x) - F(\tilde{x})\|}{\mathrm{d}(x, \tilde{x})} \qquad \forall\, x \in \mathbf{x}$$

Assuming $\mathbf{x} \sim_{\mathcal{R}} \mathbf{t}$ then enables the following conclusion for $z \in \mathbf{x} \cup \mathbf{t}$:

$$(*) \quad \mathcal{Q}_{\mathbf{x} \cup \mathbf{t}}(z) \approx \mathcal{Q}_{\mathbf{x}}(z) = \max_{\tilde{z} \in \Gamma_{\mathbf{x}}(z)} \frac{\|y(z) - y(\tilde{z})\|}{\mathrm{d}(z, \tilde{z})} = \max_{\tilde{z} \in \Gamma_{\mathbf{x}}(z)} \frac{\|\big(F(z) - y(\tilde{z})\big) + \epsilon_z\|}{\mathrm{d}(z, \tilde{z})},$$

where $\epsilon_z := y(z) - F(z)$. Notably, the rights side depends at most on one point outside $\mathbf{x}$. Therefore it includes at most one $\epsilon_z$ compared to the naive approach including both $\epsilon_z$ and $\epsilon_{\tilde{z}}$:

$$\mathcal{Q}_{\mathbf{x} \cup \mathbf{t}}(z) = \max_{\tilde{z} \in \Gamma_{\mathbf{x} \cup \mathbf{t}}(z)} \frac{\|y(z) - y(\tilde{z})\|}{\mathrm{d}(z, \tilde{z})} = \max_{\tilde{z} \in \Gamma_{\mathbf{x} \cup \mathbf{t}}(z)} \frac{\|\big(F(z) - F(\tilde{z})\big) + (\epsilon_z - \epsilon_{\tilde{z}})\|}{\mathrm{d}(z, \tilde{z})}$$

To summarize: by assuming $\mathbf{x} \sim_{\mathcal{R}} \mathbf{t}$ one can find a small $\gamma_z$ such that $\mathcal{Q}_{\mathbf{x} \cup \mathbf{t}}(z) = \mathcal{Q}_{\mathbf{x}}(z) + \gamma_z$ and trade $\epsilon_{\tilde{z}} \in \mathcal{TS}$ for $\gamma_z \in \mathbb{R}$. But whereas the first depends on the extension $F$, the latter only depends on the data.

Let now $L(F - y)$ be the Lipschitz constant of the map $F - y$ on $\mathbf{x} \cup \mathbf{t}$. Using $(*)$ and the reverse triangle inequality one can derive the following (comp. **A.4**):

$$L(F - y) \geq \max_{z \in \mathbf{x} \cup \mathbf{t}} \left| \max_{\tilde{z} \in \Gamma_{\mathbf{x} \cup \mathbf{t}}(z)} \frac{\|y(z) - y(\tilde{z})\|}{\mathrm{d}(z, \tilde{z})} - \max_{\tilde{z} \in \Gamma_{\mathbf{x} \cup \mathbf{t}}(z)} \frac{\|F(z) - F(\tilde{z})\|}{\mathrm{d}(z, \tilde{z})} \right|$$

$$= \max_{z \in \mathbf{x} \cup \mathbf{t}} \left| \max_{\tilde{z} \in \Gamma_{\mathbf{x}}(z)} \frac{\|\big(F(z) - F(\tilde{z})\big) + \epsilon_z\|}{\mathrm{d}(z, \tilde{z})} + \gamma_z - \max_{\tilde{z} \in \Gamma_{\mathbf{x} \cup \mathbf{t}}(z)} \frac{\|F(z) - F(\tilde{z})\|}{\mathrm{d}(z, \tilde{z})} \right|$$

Hence, $\epsilon_{\tilde{z}}$ and $\gamma_z$ determine a lower bound on $L(F - y)$ and decreasing it a priori may allow for finding an extension $F$ that minimizes both $\|F - y\|_{\infty}$ and $L(F - y)$ at the same time. By aligning the robustness distributions of $\mathbf{x}$ and $\mathbf{t}$ this bound will not only depend on the extension $F$, but on the data (trading the possibly uncontrollable for the controllable). Finally, the true motive for such an approach stems from functional analysis: the space of Lipschitz functions from $\mathbf{x} \cup \mathbf{t}$ to $\mathcal{TS}$, i.e. $\mathrm{Lip}(\mathbf{x} \cup \mathbf{t}, \mathcal{TS})$, is a Banach space with respect to the norm $\| \cdot \|_{\mathrm{sum}} := \| \cdot \|_{\infty} + L(\cdot)$ (comp. Cobzaş et al. (2019)). So by regarding extensions $F$ of $y$ that minimise $\|F - y\|_{\mathrm{sum}}$ we a priori restrict the size of the hypothesis space from arbitrary maps to Lipschitz maps. The completeness of $\mathrm{Lip}(\mathbf{x} \cup \mathbf{t}, \mathcal{TS})$ is of importance as it prevents sequences of extensions $(F_n)$ with $\|F_n - y\|_{\mathrm{sum}} \to 0$ to exit this smaller space[2].

---

[1] One may think of a machine learning model $F$ trained on $\mathbf{x}$ making predictions on $\mathbf{t}$.

[2] Training a machine learning model produces exactly such a sequence $F_n$.

### 3.6 SAMPLE ROBUSTNESS AND KNN

Let $F(z) := \sum_{i=1}^{\mathbf{K}} \omega_i y(z^i)$ be a $\mathbf{K}$-nearest-neighbour model with reference set $\mathbf{x}$, where $z^i$ is the i-th nearest neighbour of $z$ in $\mathbf{x}$ with weight $\omega_i$. The formula $(*)$ from **3.5** translates to:

$$\max_{\tilde{z} \in \Gamma_{\mathbf{x} \cup \mathbf{t}}(z)} \frac{\|y(z) - y(\tilde{z})\|}{\mathrm{d}(z, \tilde{z})} \quad \approx \quad \max_{\tilde{z} \in \Gamma_{\mathbf{x}}(z)} \frac{\|\sum_{i=1}^{\mathbf{K}} \omega_i y(z^i) - y(\tilde{z}) + \epsilon_z\|}{\mathrm{d}(z, \tilde{z})}$$

Hence, by assuming $\mathbf{x} \sim_{\mathcal{R}} \mathbf{t}$ we have imposed a constraint on the (deterministic) model. More precisely, it is forced to base its prediction of $z$ on those $\mathbf{K}$-samples $z^i$ close to $z$ for which the weighted linear combination of their labels suffices the above formula a priori. If $z$ is among the most robust samples, we know that a small change in feature space will only cause a small change in label space. Therefore we expect that a higher $\mathbf{K}$ produces a higher accuracy as samples close to $z$ will be more reliable predictors. Conversely, this is not the case for less robust samples as can be seen by comparing prediction and true label:

$$F(z) - y(z) = \sum_{i=1}^{\mathbf{K}} \omega_i (y(z^i) - y(z))$$

If there exists a $z^*$ close to $z$ such that the quotient $\frac{\|y(z^*) - y(z)\|}{\mathrm{d}(z^*, z)}$ is large, then by increasing $\mathbf{K}$ we are more likely to find a $z^i$ close or equal to $z^*$ causing the difference to grow. Another interesting fact is that less robust samples can by construction be considered as natural adversarial examples for a KNN model (Hendrycks et al., 2019). Assume for simplicity a classification task with $\mathbf{K}$=1: if $\mathbf{x}$ and $\mathbf{t}$ show similar robustness distributions and $x \in \mathbf{x}$ is among the less robust samples, then there likely exists a sample $t_a \in \mathbf{t}$ such that $\mathrm{d}(x, t_a)$ is small and $y(x) \neq y(t_a)$ (the true labels). However, $x$ and $t_a$ being close may cause the prediction $y(\arg\min_{z \in \mathbf{x}} \mathrm{d}(t_a, z))$ to be equal to $y(x)$.

## 4 EVALUATION

### 4.1 EVALUATION MODEL

We now determine the robustness distribution of the training- ($\mathbf{x}$) and test data ($\mathbf{t}$) for both CIFAR-10 and the Online News Popularity Dataset (short: ONP) to cover the classification and regression setup. In each case, we identify subsets of $\mathbf{x}$ and $\mathbf{t}$ stemming from the extremes of the respective robustness distribution hoping to amplify any possible effects. Then we use deterministic models (K-Nearest Neighbour=:KNN) and the non-deterministic models (Convolutional Neural Networks=:CNNs, Random Forest=:RF) to analyse performance in terms of the different subsets of $\mathbf{x}$ and $\mathbf{t}$. For this purpose, we measure accuracy and loss that indicate how much and how well a model can generalise. "Best" performances are displayed in **boldface**. The algorithms are noted in **B** as they are partially based on additional results in **A**. We also extended the analysis to include the MNIST dataset in **F**.

The CIFAR-10 dataset (Krizhevsky, 2012) consists of 60,000 $32^2$-pixel RGB-images evenly split into the classes airplane, automobile, bird, cat, deer, dog, frog, horse, ship, truck. **Figure 2** provides visual examples. The ONP dataset consists of metadata of over 39,000 articles published by Mashable (Fernandes, 2015) split into 58 predictive attributes (e.g. "number of words", "data channel", "polarity of positive/negative words",...), 2 non-predictive ("URL", "timedelta") and 1 goal field ("shares"). Although the latter is traditionally regarded as a regression dataset, one can associate a pseudo-classification task with it, where we classify popular ($\geq 1400$ shares, comp. Fernandes (2015)) and unpopular articles based on the prediction of shares. For almost all ONP-subsets, the binary labels are equally distributed (rate is 1:1).

For CIFAR-10 we used a KNN classifier and two different CNN architectures, one "small" CNN and the more complex ResNet-56 model (see **C.1** for details). For ONP, we used a KNN regressor as well as an RF regressor (see **C.2** for details). Neither for CIFAR-10 nor for ONP we used cross-validation due to the high computational effort of computing sample robustness values for each newly formed training set $\mathbf{x}$ and test set $\mathbf{t}$. One could determine $\mathcal{R}_{\mathbf{x} \cup \mathbf{t}}(x)$ for all samples beforehand and then apply cross-validation to allow for feasibility, but these robustness values would no longer be independent of each other.

For the small CNN we first estimated its overfitting threshold for the different training(sub)sets, for the ResNet-56 we measured after how many epochs validation loss did not decrease further (using callbacks). In both cases, we used $\mathbf{t}$ as the validation set (instead of held-out data) to emphasise the impact of the different robustness distributions in an artificial best-performance setup. The RF regressor we trained using different numbers of trees $\in \{1, 2, 3, 4, 5, 6, 7, 8, 9\}$ and both Mean Absolute Error (MAE) and Mean Squared Error (MSE) as loss criteria. The KNN classifier and regressor we constructed for $\mathbf{K} \in \{1, 2, 3, \ldots, 15\}$ and both uniform weights ("uni") and weights defined by Euclidean distance $\|\cdot\|_2$ ("dist"). The non-deterministic models we trained 25 times and discarded the ten highest and lowest loss- and accuracy values. Finally, we averaged the remaining five to avoid focus on individual performances. The details for each model are in **C.3**.

## 4.2 CLASSIFICATION CENTRIC ANALYSIS (CIFAR-10)

Consider the feature space $\mathcal{FS} := [0, 1]^{3 \cdot 32^2}$ with Euclidean metric $\|\cdot\|_2$ and the CIFAR-10 dataset of 50,000 training images $\mathbf{x}$ and 10,000 test images $\mathbf{t}$ labelled as $k \in \{$airplane, automobile, bird, cat, deer, dog, frog, horse, ship, truck$\}$. We regard the label maps as functions $y^k : \mathbf{x} \to \mathbb{R}$, where $y^k(x)$ is 1 if the label of $x$ is $k$ and 0 otherwise. The target space $\mathbb{R}$ is equipped with its Euclidean norm making it a Banach space. Using Algorithm 2 (comp. **B**) we determined the robustness of all samples in both $\mathbf{x}$ and $\mathbf{t}$, respectively. Afterwards we collected subsets of least- and most robust samples, the distribution of which is shown in **Table 1**. The subscript defines the subset-size in thousands (number) and whether they belong to the least- ($\mathbf{L}$) or most ($\mathbf{M}$) robust samples of $\mathbf{x}$ or $\mathbf{t}$. As an example, $\mathbf{x_{L40}}$ stands for the 40,000 least robust samples of the training set $\mathbf{x}$. The graphs in **D.1** compare the label-wise robustness distributions for $\mathbf{x}$ and $\mathbf{t}$.

Table 1: Sample distribution per label of all data(sub)sets.

| Class | $\mathbf{x}$ | $\mathbf{x_{L40}}$ | $\mathbf{x_{M40}}$ | $\mathbf{x_{L25}}$ | $\mathbf{x_{M25}}$ | $\mathbf{t}$ | $\mathbf{t_{L5}}$ | $\mathbf{t_{M5}}$ | $\mathbf{t_{L2}}$ | $\mathbf{t_{M2}}$ |
|---|---|---|---|---|---|---|---|---|---|---|
| "airplane" | 5000 | 4332 (87%) | 3538 (71%) | 3109 (62%) | 1891 | 1000 | 622 (62%) | 378 | 303 (30%) | 136 (14%) |
| "auto" | 5000 | 3171 (63%) | 4898 (98%) | 982 (20%) | 4018 | 1000 | 207 (21%) | 793 | 24 (2%) | 358 (36%) |
| "bird" | 5000 | 4540 (91%) | 3104 (62%) | 3476 (70%) | 1524 | 1000 | 675 (68%) | 325 | 366 (37%) | 116 (12%) |
| "cat" | 5000 | 3795 (76%) | 4358 (87%) | 2153 (43%) | 2847 | 1000 | 435 (44%) | 565 | 135 (14%) | 224 (14%) |
| "deer" | 5000 | 4704 (94%) | 2904 (58%) | 3844 (77%) | 1156 | 1000 | 787 (79%) | 213 | 432 (43%) | 54 (5%) |
| "dog" | 5000 | 3701 (74%) | 4403 (88%) | 2029 (41%) | 2971 | 1000 | 398 (40%) | 602 | 123 (12%) | 281 (28%) |
| "frog" | 5000 | 4544 (91%) | 3467 (69%) | 3453 (69%) | 1547 | 1000 | 711 (71%) | 289 | 312 (31%) | 94 (9%) |
| "horse" | 5000 | 3789 (76%) | 4517 (90%) | 2048 (41%) | 2952 | 1000 | 387 (39%) | 613 | 69 (7%) | 245 (25%) |
| "ship" | 5000 | 4323 (86%) | 3927 (79%) | 2921 (58%) | 2079 | 1000 | 567 (57%) | 433 | 207 (21%) | 136 (14%) |
| "truck" | 5000 | 3101 (62%) | 4884 (98%) | 985 (20%) | 4015 | 1000 | 211 (21%) | 789 | 29 (3%) | 356 (36%) |
| $\Sigma$ | 50000 | 40000 | 40000 | 25000 | 25000 | 10000 | 5000 | 5000 | 2000 | 2000 |

The relative label-distributions of both $\mathbf{x}$ and $\mathbf{t}$ are similar judging by the 50%, 20% and 80% quantiles (black percentage values are almost equal, blue and green values add up to about 100%). In **Figure. 2** some of the most- and least robust samples in $\mathbf{x}$ are displayed together with their robustness value. Here, the values of the least robust samples of "airplane" and "ship" coincide as one is the closest differently labelled image with respect to the other (and vice versa). The same holds for the least robust images of "bird" and "deer". In this classification setup, one can see that a close "visual distance" causes low sample robustness as the difference of labels is either 1 or 0. More examples are in **D.2**.

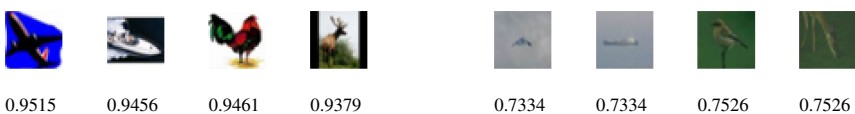

| 0.9515 | 0.9456 | 0.9461 | 0.9379 | | 0.7334 | 0.7334 | 0.7526 | 0.7526 |

Figure 2: Most- (left) and least robust samples in $\mathbf{x}$ labelled "airplane", "ship", "bird", "deer".

### 4.2.1 CNNs

The results (LOSS \ ACC) for the CNNs are displayed in **Table 2 - 4**. We noted the number of epochs it took to reach the lowest average loss on $\mathbf{t}$ (comp. **C.3**) and the corresponding performances for all training- and test(sub)sets. **Table 3** displays the performance matrix of an alternative session for the small CNN where loss was re-weighted to account for class imbalance in the subsets of $\mathbf{t}$. **D.3** displays the accuracy and loss curves of a small CNN prototype model trained on $\mathbf{x}$ for all test(sub)sets.

Table 2: Small CNN performance matrix and number of epochs until overfitting commences.

|  | $\mathbf{x}$ | $\mathbf{x_{L40}}$ | $\mathbf{x_{M40}}$ | $\mathbf{x_{L25}}$ | $\mathbf{x_{M25}}$ |
|---|---|---|---|---|---|
| Epochs | 85 | 80 | 80 | 75 | 75 |
| $\mathbf{t}$ | **0.6804** \ **0.7822** | 0.6858 \ 0.7760 | 0.6925 \ 0.7766 | 0.8123 \ 0.7428 | 0.7973 \ 0.7336 |
| $\mathbf{t_{L5}}$ | 0.6970 \ 0.7765 | **0.6819** \ **0.7772** | 0.7433 \ 0.7631 | 0.7517 \ 0.7570 | 0.8884 \ 0.7006 |
| $\mathbf{t_{M5}}$ | 0.6609 \ 0.7871 | 0.6916 \ 0.7764 | **0.6417** \ **0.7900** | 0.8808 \ 0.7276 | 0.7039 \ 0.7676 |
| $\mathbf{t_{L2}}$ | 0.6538 \ 0.7909 | **0.6358** \ **0.7953** | 0.7578 \ 0.7643 | 0.6553 \ 0.7839 | 0.9327 \ 0.6877 |
| $\mathbf{t_{M2}}$ | 0.6684 \ 0.7852 | 0.7122 \ 0.7708 | **0.6409** \ **0.7940** | 0.9742 \ 0.7166 | 0.6882 \ 0.7793 |

Table 3: Small CNN performance matrix with re-weighted LOSS (same epochs as in **Table.2**).

|  | $\mathbf{x}$ | $\mathbf{x_{L40}}$ | $\mathbf{x_{M40}}$ | $\mathbf{x_{L25}}$ | $\mathbf{x_{M25}}$ |
|---|---|---|---|---|---|
| Epochs | 85 | 80 | 80 | 75 | 75 |
| $\mathbf{t}$ | **0.6807** \ **0.7833** | 0.6912 \ 0.7771 | 0.6972 \ 0.7763 | 0.8071 \ 0.7419 | 0.7979 \ 0.7349 |
| $\mathbf{t_{L5}}$ | 0.6974 \ 0.7774 | **0.6956** \ **0.7788** | 0.7362 \ 0.7626 | 0.7651 \ 0.7563 | 0.8571 \ 0.7045 |
| $\mathbf{t_{M5}}$ | 0.7436 \ 0.7880 | 0.7658 \ 0.7761 | **0.7228** \ **0.7900** | 0.9546 \ 0.7293 | 0.8030 \ 0.7660 |
| $\mathbf{t_{L2}}$ | 0.7559 \ 0.7927 | **0.7544** \ **0.7950** | 0.8335 \ 0.7623 | 0.8102 \ 0.7816 | 0.8432 \ 0.6902 |
| $\mathbf{t_{M2}}$ | 0.7943 \ 0.7896 | 0.8634 \ 0.7688 | **0.7608** \ **0.7958** | 1.1018 \ 0.7171 | 0.8475 \ 0.7781 |

Table 4: ResNet-56 performance matrix where loss was lowest on $\mathbf{t}$ for the amount of epochs noted.

|  | $\mathbf{x}$ | $\mathbf{x_{L40}}$ | $\mathbf{x_{M40}}$ | $\mathbf{x_{L25}}$ | $\mathbf{x_{M25}}$ |
|---|---|---|---|---|---|
| Epochs | 20+1 (15) | 20+1 (13) | 15+1 (12) | 15+1 (11) | 15+1 (11) |
| $\mathbf{t}$ | **0.7568** \ **0.8535** | 0.8466 \ 0.8316 | 0.8419 \ 0.8308 | 1.0567 \ 0.7663 | 1.0567 \ 0.7656 |
| $\mathbf{t_{L5}}$ | **0.7735** \ **0.8493** | 0.8404 \ 0.8332 | 0.8811 \ 0.8186 | 1.0099 \ 0.7821 | 1.1431 \ 0.7351 |
| $\mathbf{t_{M5}}$ | **0.7397** \ **0.8585** | 0.8520 \ 0.8290 | 0.8045 \ 0.8417 | 1.1019 \ 0.7519 | 0.9715 \ 0.7938 |
| $\mathbf{t_{L2}}$ | **0.7236** \ **0.8570** | 0.7769 \ 0.8453 | 0.8691 \ 0.8168 | 0.9177 \ 0.8034 | 1.1785 \ 0.7193 |
| $\mathbf{t_{M2}}$ | **0.7542** \ **0.8561** | 0.8842 \ 0.8209 | 0.7971 \ 0.8419 | 1.1698 \ 0.7390 | 0.9576 \ 0.8010 |

**Observations (CNNs)**: performance on $\mathbf{t}$ and the number of epochs necessary to reach the least loss thereon decreased both by downsizing $\mathbf{x}$. Training on $\mathbf{x_{L40}}$ (i.e. 80% of the original training set) causes a decrease in performance on $\mathbf{t_{M5}}$ and $\mathbf{t_{M2}}$, but increases performance on $\mathbf{t_{L5}}$ and $\mathbf{t_{L2}}$. It shows the effect of training and testing on subsets which are more aligned in terms of their robustness distribution discussed in **3.5**. Similar results hold for $\mathbf{x_{M40}}$. When removing half the samples from $\mathbf{x}$, we can see the trade-off between creating more similar robustness distributions and the overall ability of the model to generalise from its training data. **Table 3** reveals furthermore that training on the more robust half of $\mathbf{x}$ causes less of a bias: loss on the test(sub)sets is more evenly distributed compared to that of models trained on $\mathbf{x_{L25}}$. This effect is likely caused by the model learning to connect small differences in feature space with large differences in label space. Hence, it upscales predictions of more robust samples that are further away from the less robust data it was trained on. For the ResNet-56, we can see the same relation between training and performing on subsets of different robustness as for the small CNN. However, there is no (partial) improvement over the baseline. We attribute this to the different learning structure of a residual neural network for which the trade-off mentioned above may behave differently.

### 4.2.2 KNN

For KNN, the highest accuracy in every case was observed for K=1[3]. **Table 5** shows the performance matrix for all data(sub)sets (note that for K = 1 both weights coincide). **D.4** displays the accuracy curves for $\mathbf{K} \in \{1, \ldots, 15\}$ and both weights using $\mathbf{x}$, $\mathbf{x_{L25}}$ and $\mathbf{x_{M25}}$ as reference sets.

Table 5: 1NN accuracy matrix.

|  | $\mathbf{x}$ | $\mathbf{x_{L40}}$ | $\mathbf{x_{M40}}$ | $\mathbf{x_{L25}}$ | $\mathbf{x_{M25}}$ |
|---|---|---|---|---|---|
| $\mathbf{t}$ | 0.3539 | 0.3434 | **0.3863** | 0.3135 | 0.3806 |
| $\mathbf{t_{L5}}$ | 0.3700 | 0.3702 | **0.3948** | 0.3656 | 0.3356 |
| $\mathbf{t_{M5}}$ | 0.3378 | 0.3166 | 0.3778 | 0.2614 | **0.4256** |
| $\mathbf{t_{L2}}$ | **0.4095** | **0.4095** | 0.3950 | **0.4095** | 0.3055 |
| $\mathbf{t_{M2}}$ | 0.3575 | 0.3155 | 0.3875 | 0.2410 | **0.4320** |

**Observations (KNN):** the highest accuracy on the most- and least robust test(sub)sets is achieved using the reference sets which are the most similar in terms of their robustness distribution, comp.

---

[3]We attribute this to the following fact: a large "closest" distance to differently labelled samples does not ensure that there exist close samples of the same label after all. This effect is amplified for image data, where changing the background does not necessarily change the label of a displayed object, but will increase Euclidean distance significantly.

**3.5**. Whereas accuracy on $t_{L2}$ did not decrease by removing more robust samples, it increased on $t_{M2}$ when removing less robust samples. This effect and the increased accuracy on $t$ by using $x_{L40}$ we attribute to the removal of natural adversarial examples as discussed in **3.6**.

### 4.3 Regression Centric Analysis (Online News Popularity Dataset)

The Online News Popularity dataset (Fernandes, 2015) was split randomly into training set $x$ and test set $t$ consisting of 32,000 and 7,644 samples, respectively. Each sample $x \in x \cup t$ is an element in $\mathcal{FS} := [0,1]^{58}$, where we use the Euclidean metric $\|\cdot\|_2$ again. The label map $y : x \to \mathbb{R}$ (where $\mathbb{R}$ is equipped with its Euclidean norm as well) assigns to each sample a normalised number of shares in $[0, 1]$. This time we used algorithm 1 (comp. **B**) to calculate the robustness for each element in $x$ and $t$ (again independently). We then defined different subsets of both $x$ and $t$ using the same notation as for the CIFAR-10 set. The sets $t_L$ and $t_M$ consist of the least- and most robust half of $t$; the sets $t_l$ and $t_m$ consist of the 500 least- and most robust samples of $t_L$ and $t_M$, respectively. **Figure 3** displays the relative distribution of shares for all data(sub)sets. The black dashed line divides samples into those below 10,000 shares on the left side and those above 10,000 shares on the right (cumulated at 110). The values in the middle display the relative amount of samples with shares above this threshold. For most of the data(sub)sets about 5% of the samples have more than 10,000 shares. For both $x_{M2}$ and $t_m$ the relative amount is about 9.4%. Conversely, while 7.6% of samples in $x_{L2}$ have more than 10.000 shares, this holds for only 0.34% of $t_l$ (i.e. less than half the relative amount). Indeed, this implicates that the label-distribution of the $\approx 6\%^4$ least robust samples in $x$ is quite different from that in $t$. **E.1** compares the overall robustness distribution of $x$ and $t$.

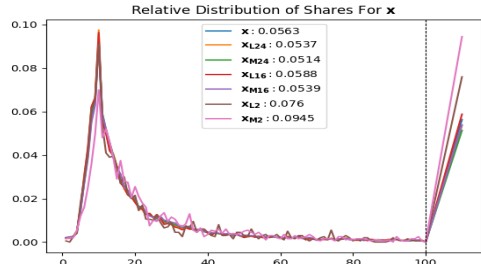 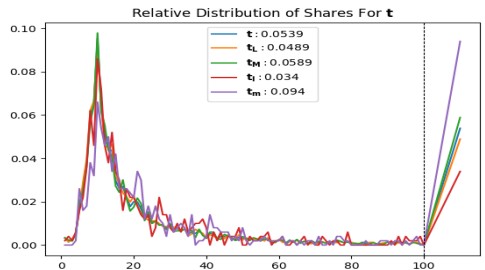

Figure 3: Relative distribution of shares for all training- and test(sub)sets.

#### 4.3.1 KNN

**Table 6** shows the performance matrix (LOSS \ ACC) for uniform ("uni")- and distance ("dist") weights, as well as the particular choice of **K** where ACC was highest on $t$. **Table 7** shows the performances and the specific **K** (in brackets) for which ACC was the highest on each test(sub)set, respectively. **E.2** displays the performance graphs.

Table 6: "uni" (left) / "dist" (right) weights where ACC was highest on $t$.

| x | $x_{L24}$ | $x_{M24}$ | k | x | $x_{L24}$ | $x_{M24}$ |
|---|---|---|---|---|---|---|
| 3 | 4 | 4 | | 3 | 4 | 4 |
| -0.1456 \ 0.5900 | **-0.1377 \ 0.5959** | -0.3945 \ 0.5907 | t | -0.1328 \ 0.5899 | **-0.1303 \ 0.5942** | -0.4141 \ 0.5917 |
| **-0.0869 \ 0.5900** | -0.0940 \ **0.6049** | -0.4290 \ 0.5942 | $t_L$ | **-0.0843 \ 0.5911** | -0.0899 \ **0.6023** | -0.4527 \ 0.5931 |
| -0.3863 \ **0.5900** | -0.3169 \ 0.5869 | **-0.2531 \ 0.5871** | $t_M$ | -0.3315 \ 0.5887 | -0.2962 \ 0.5861 | **-0.2558 \ 0.5903** |
| **0.0087 \ 0.5900** | -0.0046 \ **0.6140** | -0.1366 \ 0.6020 | $t_l$ | -0.0073 \ 0.5900 | **-0.0046 \ 0.6120** | -0.1500 \ 0.6000 |
| -0.1255 \ 0.6320 | -0.2217 \ 0.6480 | **-0.0544 \ 0.6500** | $t_m$ | -0.1222 \ 0.6280 | -0.2224 \ 0.6460 | **-0.0511 \ 0.6620** |

**Observations (KNN):** as for the KNN classifier on CIFAR-10 we can see an improvement on the most robust subsets of $t$ by choosing the corresponding subset $x_{M24}$. Conversely, this time a similar result holds for the less robust data as well. For both weights, ACC on $t_m$ is highest and drops off significantly for $t_l$. This may be explained by the different distributions of the least robust samples

---

$^4 \frac{\#x_{L2}}{\#x} = \frac{2000}{32000} = 0.0625$ and $\frac{\#t_l}{\#t} = \frac{500}{7644} = 0.0654$

Table 7: "uni" (left) / "dist" (right) weights where ACC was highest for each test(sub)set.

| x | $x_{L24}$ | $x_{M24}$ | | x | $x_{L24}$ | $x_{M24}$ |
|---|---|---|---|---|---|---|
| -0.1456 \ 0.5900 (3) | **-0.1377** \ **0.5959** (4) | -0.3945 \ 0.5907 (4) | **t** | -0.1328 \ 0.5899 (3) | **-0.1303** \ **0.5942** (4) | -0.4141 \ 0.5917 (4) |
| **-0.0867** \ 0.5947 (4) | -0.0940 \ **0.6049** (4) | -0.4290 \ 0.5942 (4) | $t_L$ | **-0.0840** \ 0.5945 (4) | -0.0899 \ **0.6023** (4) | -0.4527 \ 0.5931 (4) |
| -0.3863 \ 0.5900 (3) | **-0.2925** \ **0.5926** (5) | -0.3664 \ 0.5882 (3) | $t_M$ | -0.3315 \ 0.5887 (3) | -0.2645 \ **0.5939** (5) | **-0.2558** \ 0.5903 (4) |
| **0.0045** \ 0.6100 (4) | -0.0046 \ **0.6140** (4) | -0.1366 \ 0.6020 (4) | $t_l$ | **0.0040** \ 0.6100 (4) | -0.0046 \ **0.6120** (4) | -0.1500 \ 0.6000 (4) |
| -0.0278 \ 0.6800 (14) | -0.1309 \ 0.6720 (13) | **-0.0165** \ **0.6820** (12) | $t_m$ | -0.0239 \ 0.6800 (13) | -0.1314 \ 0.6740 (14) | **-0.0185** \ **0.6840** (12) |

in **x** and **t** (comp. $\mathbf{x_{L2}}$ and $\mathbf{t_l}$ in **Figure 3**). The optimal **K** for each test(sub)set varies between 3 and 4, except for $\mathbf{t_m}$ where it is nearly 3-4 times as high. Moreover (and in contrast to KNN on CIFAR-10), this causes an increase in accuracy of about 2% -5% in accordance with our theoretic explanation in **3.6**.

### 4.3.2 RANDOM FOREST

**Table 8** shows the performances matrix (LOSS \ ACC) and the number of trees where ACC on **t** was the highest. **Table 9** shows the performances and numbers of trees (in brackets) for which ACC was the highest on each test(sub)set, respectively. **E.3** displays the performance graphs.

Table 8: MAE (left) / MSE (right) as loss-criteria where ACC was highest on **t**.

| x | $x_{L24}$ | $x_{M24}$ | | x | $x_{L24}$ | $x_{M24}$ |
|---|---|---|---|---|---|---|
| 5 | 4 | 3 | Trees | 7 | 6 | 4 |
| -0.1867 \ **0.5951** | -0.2884 \ 0.5907 | **-0.1760** \ 0.5920 | **t** | **-0.1148** \ **0.5953** | -0.2796 \ 0.5909 | -0.1305 \ 0.5928 |
| -0.1176 \ 0.5968 | -0.1583 \ 0.5964 | **-0.1006** \ **0.5974** | $t_L$ | **-0.0590** \ **0.5998** | -0.1116 \ 0.5959 | -0.0616 \ 0.5968 |
| **-0.4461** \ **0.5922** | -0.6904 \ 0.5848 | -0.4743 \ 0.5864 | $t_M$ | -0.3814 \ **0.5905** | -0.9062 \ 0.5873 | -0.3704 \ 0.5884 |
| 0.0147 \ 0.6256 | -0.0144 \ 0.6232 | **0.0236** \ 0.6232 | $t_l$ | **0.0194** \ **0.6296** | -0.0013 \ 0.6276 | 0.0138 \ 0.6256 |
| **-0.3125** \ **0.6564** | -0.4946 \ 0.6376 | -0.3448 \ 0.6408 | $t_m$ | -0.3278 \ **0.6616** | -0.7771 \ 0.6456 | -0.3306 \ 0.6452 |

Table 9: MAE (left) / MSE (right) as loss-criteria where ACC on each test(sub)set is highest.

| x | $x_{L24}$ | $x_{M24}$ | | x | $x_{L24}$ | $x_{M24}$ |
|---|---|---|---|---|---|---|
| -0.1867 \ **0.5951** (5) | -0.2884 \ 0.5907 (4) | **-0.1760** \ 0.5920 (3) | **t** | **-0.1148** \ **0.5953** (7) | -0.2796 \ 0.5909 (6) | -0.1305 \ 0.5928 (4) |
| -0.2098 \ 0.5973 (3) | -0.1583 \ 0.5964 (4) | **-0.1006** \ **0.5974** (3) | $t_L$ | **-0.0348** \ **0.6006** (9) | -0.1617 \ 0.5963 (4) | -0.0616 \ 0.5968 (4) |
| -0.4461 \ **0.5922** (5) | -0.8000 \ 0.5867 (3) | **-0.3172** \ 0.5899 (4) | $t_M$ | -0.5619 \ **0.5925** (4) | -0.9062 \ 0.5873 (6) | **-0.2197** \ 0.5895 (7) |
| 0.0065 \ **0.6316** (3) | -0.0198 \ 0.6288 (3) | **0.0199** \ 0.6244 (5) | $t_l$ | -0.0003 \ 0.6352 (6) | -0.0034 \ **0.6356** (7) | **0.0172** \ 0.6288 (5) |
| -0.2920 \ **0.6620** (7) | -0.2708 \ 0.6512 (9) | **-0.2054** \ **0.6620** (8) | $t_m$ | -0.3217 \ **0.6692** (8) | -0.6496 \ 0.6520 (9) | **-0.2261** \ 0.6672 (8) |

**Observations (RF):** using 75% of the original training data correlates with a a slight decrease in performance (independent of the metric) and a smaller optimal number of trees for $\mathbf{x_{M24}}$ than for $\mathbf{x_{L24}}$. When training on $\mathbf{x_{L24}}$, LOSS on $\mathbf{t_M}$ and $\mathbf{t_m}$ increases significantly. This is likely caused by the model learning to connect small changes in feature space with large changes in label space. Hence, it tends to overshoot predictions for more robust data, which is also in accordance with our observations in **Table 3**. As for KNN, we can see two things: (i) a large difference in ACC on $\mathbf{t_l}$ and $\mathbf{t_m}$ and (ii) the optimal hyperparameter being significantly higher for $\mathbf{t_m}$. The first we attribute again to the different distributions of $\mathbf{x_{L2}}$ and $\mathbf{t_l}$; for the second we expect a similar explanation as for KNN (comp. **3.6**).

## 5 CONCLUSION AND FUTURE WORK

We introduced the concept of sample robustness for measuring how sensitive elements of a dataset are towards label-changing perturbations. We provided a theoretical motivation and analysed the robustness distribution of different datasets, as well as the connection to model performance. In concordance with our theoretical analysis, we found that it is possible to boost performance on specific test(sub)sets by choosing training(sub)sets exhibiting similar robustness distributions. Empirical results, however, indicate that there is a model-dependent trade-off between discarding samples to align these distributions and the general ability of a model to generalise from its training- or reference data. Finally, we found that optimal hyperparameters may also depend on the robustness of both the training- and test set. Possible future research directions are: **(i)** expanding experiments (using cross-validation, more datasets and models, different metrics, etc.); **(ii)** analyse optimal model-hyperparameters as functions $h(\mathbf{x_{AB}}, \mathbf{t_{AB}})$ in terms of training- ($\mathbf{x_{AB}}$) and test(sub)sets ($\mathbf{t_{AB}}$) of different parts of the robustness spectrum; **(iii)** explore the relation between training on more- or less robust data and model-robustness, e.g. susceptibility to adversarial examples (Szegedy et al., 2013).

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

# A  ADDITIONAL THEORY AND PROOFS:

## A.1  PROPOSITION:

Let $L(y)$ be the Lipschitz constant of the label map $y : \mathbf{x} \to \mathcal{TS}$. The following equality holds:

$$\max_{x \in \mathbf{x}} \mathcal{Q}_{\mathbf{x}}(x) = L(y)$$

*Proof.* One has $L(y) \geq \max_{x \in \mathbf{x}} \mathcal{Q}_{\mathbf{x}}(x)$ by construction. Furthermore, as $\mathbf{x}$ is compact:

$$L(y) = \max_{x \neq \tilde{x}} \frac{\|y(x) - y(\tilde{x})\|}{\mathrm{d}(x, \tilde{x})} = \frac{\|y(x_0) - y(\tilde{x}_0)\|}{\mathrm{d}(x_0, \tilde{x}_0)}$$

for some $x_0 \neq \tilde{x}_0$ such that $y(x_0) \neq y(\tilde{x}_0)$. Thus:

$$\frac{\|y(x_0) - y(\tilde{x}_0)\|}{\mathrm{d}(x_0, \tilde{x}_0)} \leq \mathcal{Q}_{\mathbf{x}}(x_0) \leq \max_{x \in \mathbf{x}} \mathcal{Q}_{\mathbf{x}}(x)$$

$\square$

## A.2  PROPOSITION:

$\mathcal{R}_{\mathbf{x}}(x)$ is independent of rescaling the label map $y$.

*Proof.* Suppose we have a label map $\tilde{y} := ay$ for $a \in \mathbb{R} - \{0\}$, then:

$$\tilde{\mathcal{R}}_{\mathbf{x}}(x) := \frac{\|\tilde{y}\|_\infty}{\displaystyle\max_{\tilde{x} \in \Gamma_{\mathbf{x}}(x)} \frac{\|\tilde{y}(x) - \tilde{y}(\tilde{x})\|}{\mathrm{d}(x, \tilde{x})} + \|\tilde{y}\|_\infty}$$

$$= \frac{|a| \, \|y\|_\infty}{|a| \displaystyle\max_{\tilde{x} \in \Gamma_{\mathbf{x}}(x)} \frac{\|y(x) - y(\tilde{x})\|}{\mathrm{d}(x, \tilde{x})} + |a| \, \|y\|_\infty} = \mathcal{R}_{\mathbf{x}}(x)$$

$\square$

## A.3  PROPOSITION:

Let $x \in \mathbf{x}$ with reach $\mathbf{r}(x)$. If $y(x) \in \{0, e\}$ for all $x \in \mathbf{x}$, where $e$ lies on the unit sphere in $\mathcal{TS}$ (i.e. $y$ is a binary label map), then:

$$\mathcal{R}_{\mathbf{x}}(x) = \frac{\mathbf{r}_{\mathbf{x}}(x)}{\mathbf{r}_{\mathbf{x}}(x) + 1}$$

*Proof.* It holds that:

$$\mathcal{R}_{\mathbf{x}}(x) = \left( \max_{\tilde{x} \in \Gamma_{\mathbf{x}}(x)} \frac{\|y(x) - y(\tilde{x})\|}{\mathrm{d}(x, \tilde{x})} + 1 \right)^{-1} = \left( \max_{\tilde{x} \in \Gamma_{\mathbf{x}}(x)} \frac{1}{\mathrm{d}(x, \tilde{x})} + 1 \right)^{-1}$$

$$= \left( \frac{1}{\displaystyle\min_{\tilde{x} \in \Gamma_{\mathbf{x}}(x)} \mathrm{d}(x, \tilde{x})} + 1 \right)^{-1} = \left( \frac{1}{\mathbf{r}_{\mathbf{x}}(x)} + 1 \right)^{-1} = \left( \frac{1 + \mathbf{r}_{\mathbf{x}}(x)}{\mathbf{r}_{\mathbf{x}}(x)} \right)^{-1}$$

$\square$

## A.4 LEMMA:

With the notions of **3.5** it holds:

$$L(F - y) \geq \max_{z \in \mathbf{x} \cup \mathbf{t}} \left| \max_{\tilde{z} \in \Gamma_{\mathbf{x} \cup \mathbf{t}}(z)} \frac{\|y(z) - y(\tilde{z})\|}{\mathrm{d}(z, \tilde{z})} - \max_{\tilde{z} \in \Gamma_{\mathbf{x} \cup \mathbf{t}}(z)} \frac{\|F(z) - F(\tilde{z})\|}{\mathrm{d}(z, \tilde{z})} \right|$$

*Proof.* Using the reverse triangle inequality twice:

$$\left| \max_{\tilde{z} \in \Gamma_{\mathbf{x} \cup \mathbf{t}}(z)} \frac{\|y(z) - y(\tilde{z})\|}{\mathrm{d}(z, \tilde{z})} - \max_{\tilde{z} \in \Gamma_{\mathbf{x} \cup \mathbf{t}}(z)} \frac{\|F(z) - F(\tilde{z})\|}{\mathrm{d}(z, \tilde{z})} \right|$$

$$\leq \max_{\tilde{z} \in \Gamma_{\mathbf{x} \cup \mathbf{t}}(z)} \left| \frac{\|y(z) - y(\tilde{z})\|}{\mathrm{d}(z, \tilde{z})} - \frac{\|F(z) - F(\tilde{z})\|}{\mathrm{d}(z, \tilde{z})} \right|$$

$$\leq \max_{\tilde{z} \in \Gamma_{\mathbf{x} \cup \mathbf{t}}(z)} \frac{\|y(z) - y(\tilde{z}) - F(z) + F(\tilde{z})\|}{\mathrm{d}(z, \tilde{z})}$$

To see the first inequality, let $X_z := \Gamma_{\mathbf{x} \cup \mathbf{t}}(z)$, $V_z(\tilde{z}) := \frac{\|y(z) - y(\tilde{z})\|}{\mathrm{d}(z, \tilde{z})}$, $W_z(\tilde{z}) := \frac{\|F(z) - F(\tilde{z})\|}{\mathrm{d}(z, \tilde{z})}$ and note that $V = |V|$ and $W = |W|$. Then, by using the uniform norm on $X_z$ defined as

$$\|V\|_{\infty, X_z} := \max_{\tilde{z} \in X_z} |V(\tilde{z})|,$$

we see that the first inequality is indeed a consequence of the reverse triangle inequality:

$$\left| \|V\|_{\infty, X_z} - \|W\|_{\infty, X_z} \right| \leq \|V - W\|_{\infty, X_z}$$

Taking the maximum over $z \in \mathbf{x} \cup \mathbf{t}$ yields the statement. $\qquad\square$

## B ALGORITHM (SAMPLE ROBUSTNESS)

Let $\mathbf{x} \subset \mathcal{FS}$ be a dataset in some feature space with metric $\mathrm{d}$ and label map $y$ taking values in some target space $\mathcal{TS}$ with norm $\|\cdot\|$. For a sample $x \in \mathcal{FS}$ with label $y(x)$ such that $\|y(x)\| \leq \max_{\tilde{x} \in \mathbf{x}} \|y(\tilde{x})\|$ one can calculate its robustness using the algorithms below. Note that the second algorithm is a special case of the first.

---

**Algorithm 1** Calculate $\mathcal{R}_{\mathbf{x}}(x)$ (General $y$)

---

**Require:** $\#y(\mathbf{x}) \geq 2, \|y\|_\infty = 1$
  $y \leftarrow y(x)$
  $y_j \leftarrow y(x_j)$ for any $x_j \in \mathbf{x}$ s.t. $y_j \neq y$
  $Q \leftarrow \dfrac{\|y - y_j\|}{\mathrm{d}(x, x_j)}$
  **for** $x_i \in \mathbf{x}$ **do**
    $y_i \leftarrow y(x_i)$
    **if** $y \neq y_i$ **then**
      $Q \leftarrow \max\{Q, \dfrac{\|y - y_i\|}{\mathrm{d}(x, x_i)}\}$
    **end if**
  **end for**
  **print** $(Q + 1)^{-1}$

---

---

**Algorithm 2** Calculate $\mathcal{R}_\mathbf{x}(x)$ (Binary $y$)

---

**Require:** $y(\mathbf{x}) = \{0, e\}$ with $\|e\| = 1$
  $y \leftarrow y(x)$
  $r \leftarrow \max_{\mathbf{x}}$
  **for** $x_i \in \mathbf{x}$ **do**
    $y_i \leftarrow y(x_i)$
    **if** $y \neq y_i$ **then**
      $r \leftarrow \min\{r, \mathrm{d}(x, x_i)\}$
    **end if**
  **end for**
  **print** $\dfrac{r}{r + 1}$

---

## C  MODELS

### C.1  CIFAR-10

#### SMALL CNN

The small CNN architecture was taken initially from the Keras homepage[5] and consists of two convolutional blocks, each built using two convolutional layers with kernel size (3,3) and RELU activation function, followed by MaxPooling with pool size (2,2) and Dropout ratio of 0.25. After this, there are two dense layers (512/RELU and 10/Softmax) with a Dropout ratio of 0.5 in between. It was compiled using the RMSpropo optimizer with learning rate $10^{-4}$ and decay of $10^{-6}$. We used the categorical cross-entropy as our loss function and trained the model with a batch size of 32.

#### RESNET-56

The ResNet-56 architecture (He et al., 2016) was also taken directly from the Keras homepage[6]. We used n=9, batch sizes of 32 and the pre-defined learning rates of $10^{-3}$ and $10^{-4}$.

#### KNN

For the KNN classifier we used the scikit-learn pre-defined algorithm[7] with both uniform weights ("uni") and weights defined by the Euclidean metric $\| \cdot \|_2$ ("dist").

### C.2  ONLINE NEWS POPULARITY DATA SET (ONP)

#### KNN

For the KNN regressor we used the scikit-learn pre-defined algorithm[8] with both uniform weights ("uni") and weights defined by the Euclidean metric $\| \cdot \|_2$ ("dist"). The loss function is defined as

$$\text{LOSS} := 1 - \frac{\sum_i (y_i^{true} - y_i^{pred})^2}{\sum_i \left(y_i^{true} - \text{MEAN}(y^{true})\right)^2} \in (-\infty, 1],$$

where 1 is the optimum. Accuracy is based on the pseudo-classification task (see 4.1).

---

[5] It was recently removed.
[6] https://keras.io/zh/examples/cifar10_resnet/
[7] https://scikit-learn.org/stable/modules/generated/sklearn.neighbors.KNeighborsClassifier.html
[8] https://scikit-learn.org/stable/modules/generated/sklearn.neighbors.KNeighborsRegressor.html

RANDOM FOREST

To construct a random forest regressor we also used the scikit-learn pre-defined algorithm[9]. We measured the same loss as for KNN. Again, accuracy is based on the pseudo classification task (see 4.1).

## C.3 PROCEDURE DETAILS FOR THE NON-DETERMINISITC MODELS

SMALL CNN

For the small CNN, we trained multiple models for an increasing amount of epochs $e$ until we could roughly pinpoint the overfitting-threshold $e_{over}$ on the test set $\mathbf{t}$ for every training(sub)set. Then, in a second session, we trained 25 independent models (always starting with different weights) for $e_{over}$ epochs and collected their performances on all of the test(sub)sets $\mathbf{t}, \mathbf{t_{L5}}, \mathbf{t_{M5}}, \mathbf{t_{L2}}, \mathbf{t_{M2}}$ in five accuracy- and five loss-lists, respectively. From each list, we discarded the ten highest and the ten lowest values of the 25 and then averaged the remaining five in order to avoid focus on individual model performance. After this we repeated the procedure for $e_k := e_{over} \pm 5k$, $k \in \mathbb{Z}$, epochs until we found a local loss-minimum on $\mathbf{t}$ away from the lowest and highest number of epochs in each case. However, it is still possible to fall victim to the stochastic behaviour of neural networks, and in some cases, we repeated second sessions to account for this fact. The optimal number of epochs may vary in a $\pm 5$ vicinity. Also, since double-descent has been discovered (Nakkiran et al., 2020), one may always question the optimal overfitting threshold.

RESNET-56

The ResNet-56 models were trained 15 times on each of the different training(sub)sets for 80 epochs using a learning rate of $10^{-3}$. Here, we used callbacks to save the best weights and to identify the particular number of epochs for which loss on $\mathbf{t}$ was the lowest. We then picked an individual epoch-threshold from $\{5k \mid k \in \mathbb{N}\}$ such that at most 3 of the 15 values lay above it (to account for outliers). Finally, we conducted second sessions similar to those for the small CNN, where we (i) trained the models for the number of epochs indicated by these upper thresholds, (ii) saved the best weights, (iii) rebuilt the models with these weights and (iv) topped everything off with an additional epoch of training using a lower learning rate of $10^{-4}$.

The epoch-history throughout training the 15 models per training(sub)set is shown below. The number in the brackets is the rounded average of the 15 values after we discarded the highest- and lowest three entries.

- $\mathbf{x}$: [13, 17, 20, 12, 12, 15, 15, 18, 13, 19, 12, 23, 23, 17, 8 ]   (15)

- $\mathbf{x_{L40}}$: [ 9, 17, 14, 15, 12, 18, 13, 16, 11, 12, 64, 7, 13, 12, 9 ]   (13)

- $\mathbf{x_{M40}}$: [14, 11, 21, 11, 9, 10, 77, 15, 11, 14, 10, 8, 14, 8, 13 ]   (12)

- $\mathbf{x_{L25}}$: [12, 7, 9, 14, 12, 77, 9, 14, 12, 21, 5, 7, 14, 7, 11 ]   (11)

- $\mathbf{x_{M25}}$: [ 6, 15, 9, 13, 13, 9, 7, 10, 15, 11, 13, 17, 10, 10, 9 ]   (11)

RANDOM FOREST

For every number of trees, we simply used the same second session procedure as for the small CNN, tailored for the ONP training- and test(sub)sets.

---

[9]https://scikit-learn.org/stable/modules/generated/sklearn.ensemble.RandomForestRegressor.html

# D    CIFAR-10: ADDITIONAL MATERIAL

## D.1    LABEL-WISE ROBUSTNESS DISTRIBUTION

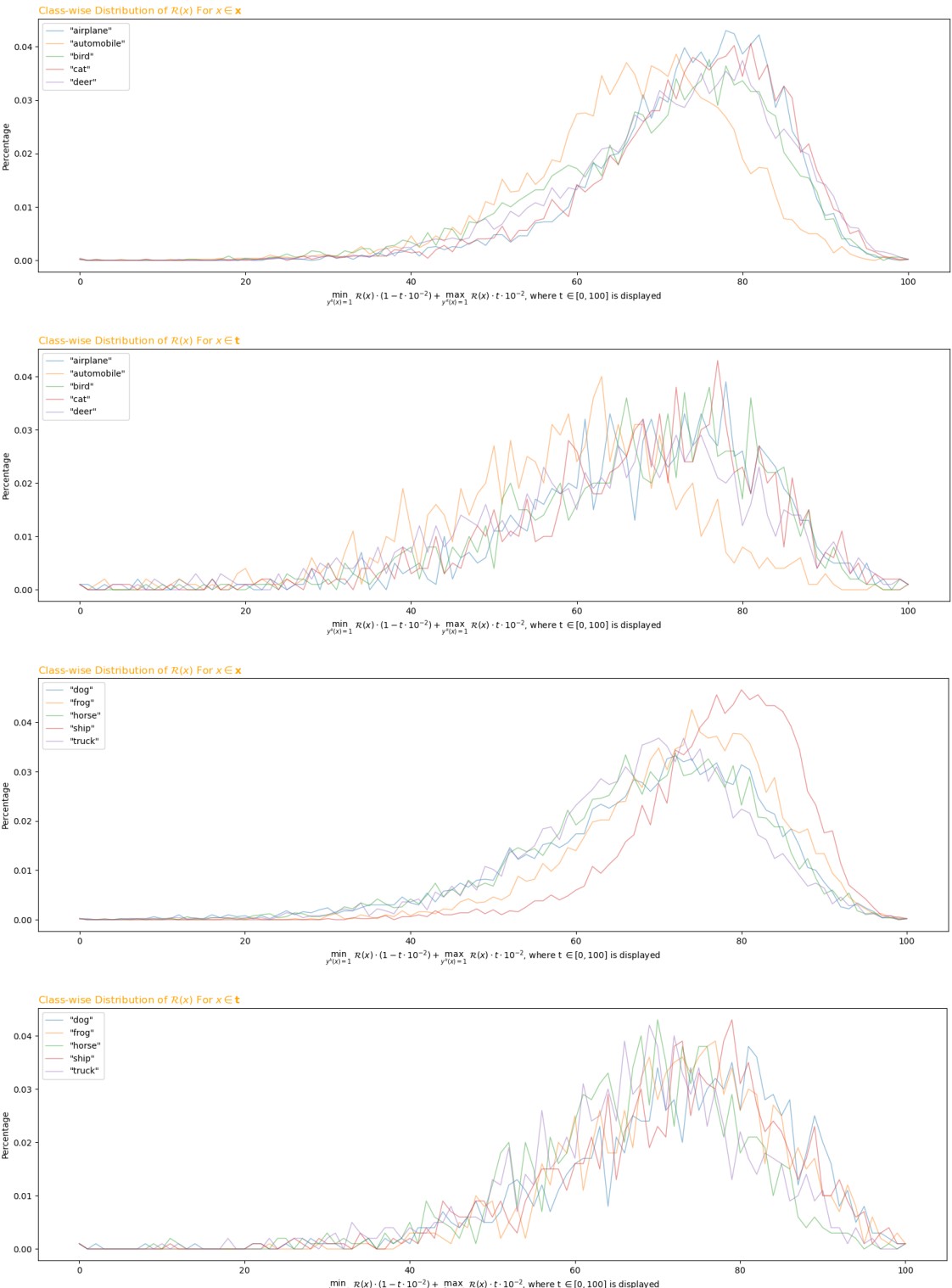

Figure 4: Relative class-wise sample robustness distribution for **x** and **t**.

### D.2    LEAST- AND MOST ROBUST SAMPLES IN THE CIFAR-10 TRAININGSET

Below is the juxtaposition of the least- and most robust samples in **x** for each class together with
their respective sample robustness and the average model output probability (in the brackets) of each
image representing its respective class. Here, the average is determined using the small CNN and the
same second session procedure described in **C.3**. Notably, the background of each animal or object
displayed in the image impacts robustness as differently coloured surroundings will significantly
increase distance w.r.t. $\|\cdot\|_2$. Comparing the robustness and average model probability for the
"bird"- and "frog" images, there seems to be no definitive relationship between those values.

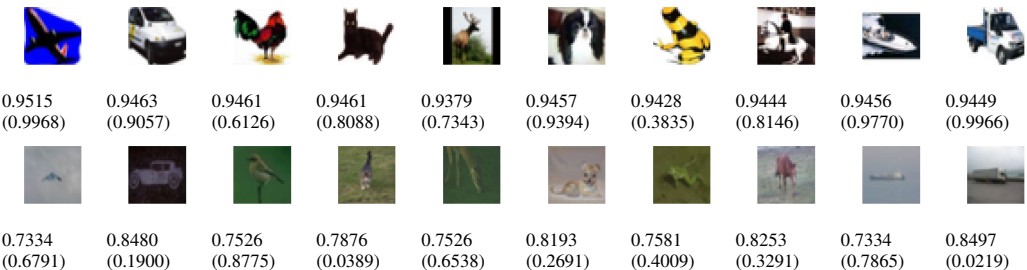

| 0.9515 | 0.9463 | 0.9461 | 0.9461 | 0.9379 | 0.9457 | 0.9428 | 0.9444 | 0.9456 | 0.9449 |
| (0.9968) | (0.9057) | (0.6126) | (0.8088) | (0.7343) | (0.9394) | (0.3835) | (0.8146) | (0.9770) | (0.9966) |

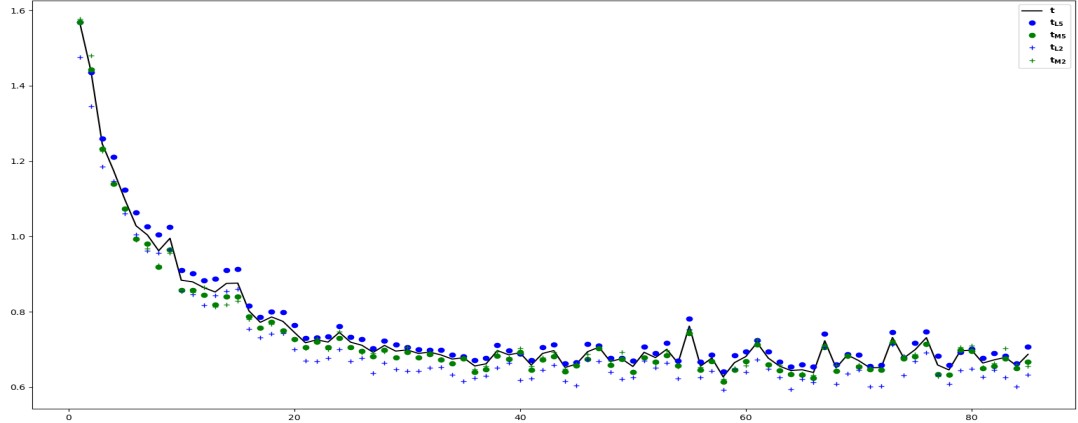

| 0.7334 | 0.8480 | 0.7526 | 0.7876 | 0.7526 | 0.8193 | 0.7581 | 0.8253 | 0.7334 | 0.8497 |
| (0.6791) | (0.1900) | (0.8775) | (0.0389) | (0.6538) | (0.2691) | (0.4009) | (0.3291) | (0.7865) | (0.0219) |

Figure 5: Most- (upper row) and least robust samples (lower row) in the CIFAR-10 training set.

### D.3    PROTOTYPE MODEL LOSS AND ACCURACY OVER 85 EPOCHS

We trained a single small CNN (see C.1) on **x** for 85 epochs (which approximately showed the
average performance in **Table 2**) and monitored its performance throughout the process. The graphs
below show the loss and accuracy values for all test(sub)sets. From the beginning, the model did
perform better on the more robust half of the test set $\mathbf{t_{M5}}$ than on the less robust half $\mathbf{t_{L5}}$. Loss
on $\mathbf{t_{L2}}$ is almost always the least of all five, though this does not necessarily cause a high accuracy.
Indeed, the subset on which the model has the highest accuracy changes every few epochs from $\mathbf{t_{L2}}$
to $\mathbf{t_{M2}}$ and vice versa throughout the whole training process.

Figure 6: Epoch-wise learning behaviour of a small CNN trained on the CIFAR-10 training set **x**
approximately expressing the loss in **Table 2**.

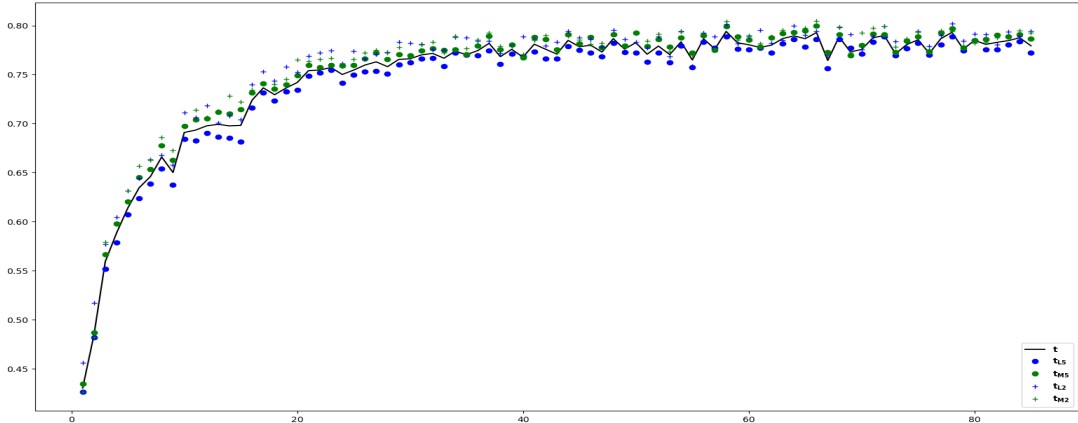

Figure 7: Epoch-wise learning behaviour of a small CNN trained on the CIFAR-10 training set **x** approximately expressing the accuracy in **Table 2**.

## D.4 KNN

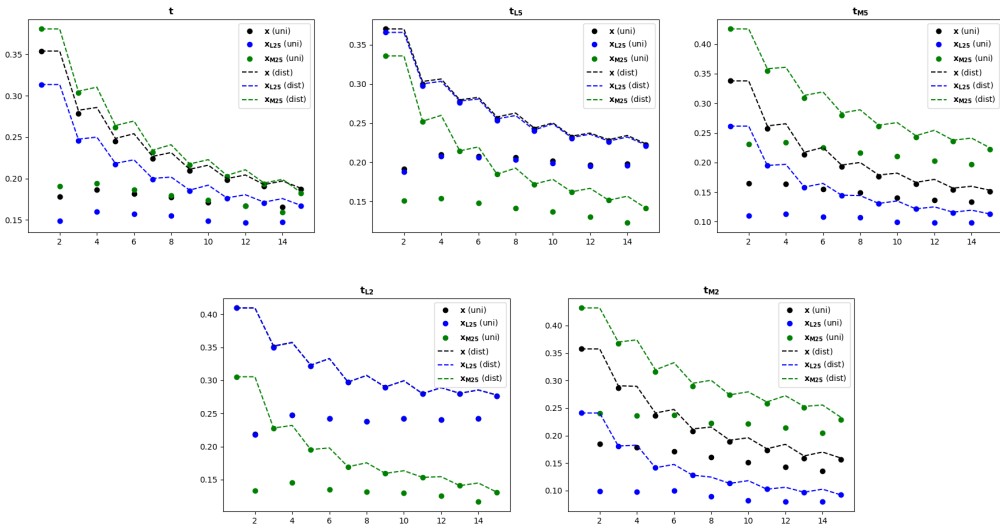

Figure 8: KNN for CIFAR-10, split for each test(sub)set. The colour of the graph corresponds to the reference training set.

# E ONP: ADDITIONAL MATERIAL

## E.1 RELATIVE ROBUSTNESS DISTRIBUTION

The minimum- and maximum sample robustness for **x** and **t** are:

$$\min_{x \in \mathbf{x}} \mathcal{R}(x) = 0.2607, \quad \max_{x \in \mathbf{x}} \mathcal{R}(x) = 0.7616, \quad \min_{x \in \mathbf{t}} \mathcal{R}(x) = 0.3792, \quad \max_{x \in \mathbf{t}} \mathcal{R}(x) = 0.8195$$

**Figure 9** displays the relative sample robustness distributions. Overall, they exhibit very similar behaviour (the peak for **x** is slightly higher).

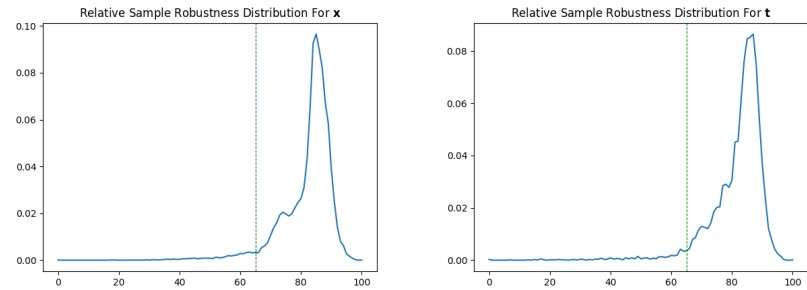

Figure 9: Relative robustness distributions for $\mathbf{x}$ and $\mathbf{t}$. The green dashed line marks the 65% threshold. For $\mathbf{x}$, 1272 samples are below this robustness level ($\approx 4\%$); for $\mathbf{t}$, 267 ($\approx 3.5\%$).

## E.2 KNN

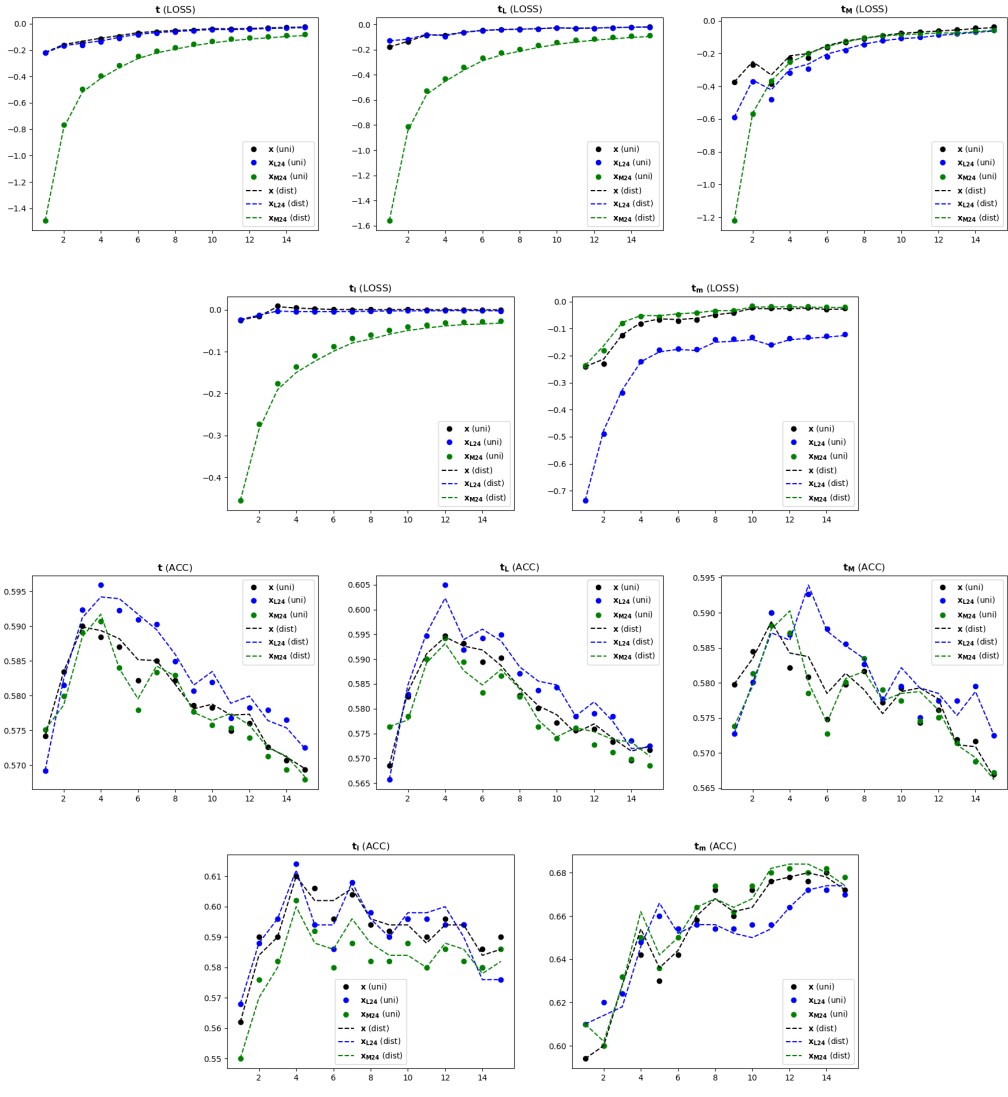

Figure 10: KNN LOSS and ACC for ONP, split for each test(sub)set. The colour of the graph corresponds to the reference set.

## E.3 RANDOM FOREST

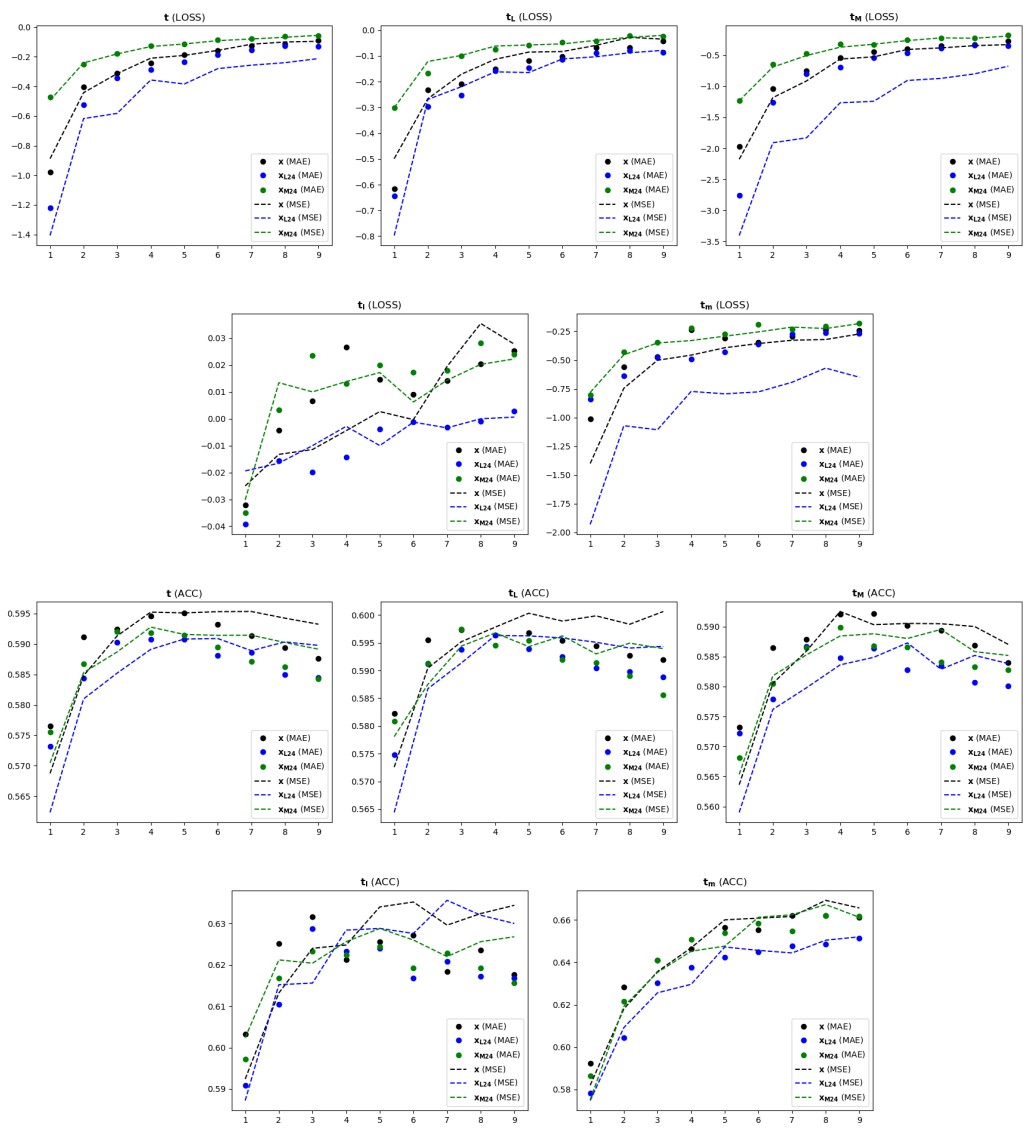

Figure 11: RF LOSS and ACC for ONP, split for each test(sub)set. The colour of the graph corresponds to the reference training set.

## F MNIST

The MNIST data set (Lecun et al., 1998) consists of 60,000 training- ($\mathbf{x}$) and 10,000 test- ($\mathbf{t}$) $28^2$-pixel greyscale images of handwritten digits (we used the same mathematical notions as for the CIFAR-10 set). **Table 10** shows the label-distributions of different data(sub)sets (again, the notation is the same as for the CIFAR-10 set). The relative label-distributions of both $\mathbf{x}$ and $\mathbf{t}$ are similar judging by the 50%, 20% and 80% quantiles (black percentage values are almost equal, blue and green values add up to about 100% except for class "1"). Overall, images displaying the digits "1", "4", "7" and "9" were very prominent in $\mathbf{x_{L48}}, \mathbf{x_{L30}}, \mathbf{t_{L5}}, \mathbf{t_{L2}}$, whereas images in $\mathbf{x_{M48}}, \mathbf{x_{M30}}, \mathbf{t_{M5}}, \mathbf{t_{M2}}$ mainly consist of "0"s, "2"s and "6"s.

Table 10: Sample distribution per label of all data(sub)sets.

| Digit | x | $x_{L48}$ | $x_{M48}$ | $x_{L30}$ | $x_{M30}$ | t | $t_{L5}$ | $t_{M5}$ | $t_{L2}$ | $t_{M2}$ |
|---|---|---|---|---|---|---|---|---|---|---|
| "0" | 5923 | 2632 (44%) | 5853 (99%) | 632 (11%) | 5291 | 980 | 94 (10%) | 886 | 2 (0%) | 562 (57%) |
| "1" | 6742 | 6739 (100%) | 2760 (41%) | 6702 (99%) | 40 | 1135 | 1125 (99%) | 10 | 896 (79%) | 1 (0%) |
| "2" | 5958 | 2931 (49%) | 5813 (98%) | 815 (14%) | 5143 | 1032 | 176 (17%) | 856 | 14 (1%) | 471 (46%) |
| "3" | 6131 | 5040 (82%) | 5828 (95%) | 2219 (36%) | 3912 | 1010 | 393 (39%) | 617 | 34 (3%) | 161 (16%) |
| "4" | 5842 | 5333 (91%) | 4310 (74%) | 3819 (65%) | 2023 | 982 | 656 (67%) | 326 | 236 (24%) | 91 (9%) |
| "5" | 5421 | 4567 (84%) | 5164 (95%) | 1895 (35%) | 3526 | 892 | 292 (33%) | 600 | 18 (2%) | 150 (17%) |
| "6" | 5918 | 4285 (72%) | 5504 (93%) | 2004 (34%) | 3914 | 958 | 265 (28%) | 693 | 18 (2%) | 331 (35%) |
| "7" | 6265 | 5862 (94%) | 4257 (68%) | 4566 (73%) | 1699 | 1028 | 743 (72%) | 285 | 289 (28%) | 47 (5%) |
| "8" | 5851 | 4832 (83%) | 5577 (95%) | 2185 (37%) | 3666 | 974 | 387 (40%) | 587 | 22 (2%) | 155 (16%) |
| "9" | 5949 | 5779 (97%) | 2934 (49%) | 5163 (87%) | 786 | 1009 | 869 (86%) | 140 | 471 (47%) | 31 (3%) |
| Σ | 60000 | 48000 | 48000 | 30000 | 30000 | 10000 | 5000 | 5000 | 2000 | 2000 |

The authors of Papernot et al. (2016a) noticed that when crafting source-target pairs of adversarial examples from the MNIST test set, "classes "0", "2" and "8" are hard to start with, while classes "1", "7" and "9" are easy to start with". Interestingly, comparing the compositions of the 2000 least- and most robust samples we can see that the three most prominent classes in $t_{L2}$ were "1", "7" and "9" (and close thereafter "4"), while the two most prominent classes in $x_{M2}$ were "0" and "2" (followed by "6" and "3" before "8").

**Figure 12** displays the most and least robust samples of each class together with their sample robustness and the average probability of a shallow CNN with an average accuracy of about 98% (25 models were trained independently and accuracy values were collected similarly to the second session for the small CNN on CIFAR-10). The boldness of the written digit has a beneficial impact on its robustness as it will increase Euclidean distance significantly. We can also see that the least robust image of a "3" presents itself as a mislabelled "9". Indeed, this makes sense as distance in feature space to another image of a "9" in x is relatively small.

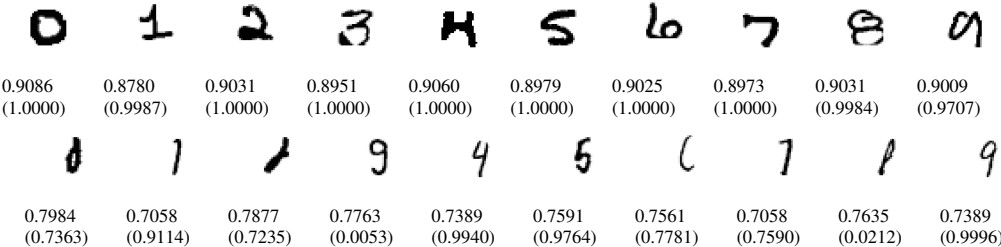

Figure 12: Most- (upper row) and least robust (lower row) samples of each label in the MNIST training set ("0" - "9" from left to right).

One may be tempted to regard sample robustness as a tool of anomaly detection (Chalapathy & Chawla, 2019; Beggel et al., 2019) as it rightfully identifies the mislabelled image of a "9". To underline that it is an utterly intrinsic and metric-dependent concept, consider the set of "handwritten digits" in **Figure 13** and their respective robustness values (note that the image of the "4" is not missing but plain white). The relatively high value of the "3" being a "4" may be caused by the missing normalization of size and position (as was done for the MNIST set).

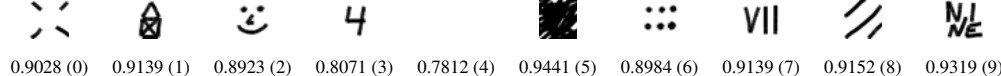

0.9028 (0)   0.9139 (1)   0.8923 (2)   0.8071 (3)   0.7812 (4)   0.9441 (5)   0.8984 (6)   0.9139 (7)   0.9152 (8)   0.9319 (9)

Figure 13: Robust samples of "handwritten digits" labelled as the number in the brackets.

