# OpenReview forum: "Introducing Sample Robustness"
_ICLR.cc/2021/Conference — Reject_

### Official Review · AnonReviewer3 · 2020-10-27
**Extensive analysis, but concerns about their relevance and value beyond existing work.**

**Rating:** 4
**Confidence:** 3

**Review:**

Overview:

The authors introduce sample robustness, a pointwise measure of the sensitivity of the label map to perturbations in the feature space. They do so by taking the pointwise Lipschitz constant of the label map and normalizing it by the label norm. The authors show that training on samples that are / are not sensitive helps on a test set that is / is not sensitive.

Positives:

The authors do extensive experimentation to evaluate the effect of training with various data subsets constructed by taking the most or least sensitive samples, and testing on other subsets of sensitive / not sensitive samples.

The results generally show that performance on non-sensitive subsets improves when training with non-sensitive data.

Negatives:

I'm not terribly convinced by the experimental results. It seems like the conclusion of the paper is that if the goal is to get good performance on the standard test set, you should just use the original data without taking into account sample robustness.

There are gains for specific subsets, but the paper doesn't really do a good job of convincing me why I should care about this. It seems like the end goal should be performance on whatever test set I originally collected, and not a subset where you remove particular hard or easy examples. This concern is exacerbated by the fact that the train and test data selection procedures are matched (i.e. you select points based on sample robustness and evaluate on similar selection criteria).

Finally, I'm not sure if this notion of sample robustness is conceptually novel compared to methods like adversarial pruning (Yang 2019). The authors could have done more in linking the effects of sample robustness to the margin or generalization properties of the resulting classifier. For an example of such discussion see the recent line of work on r-consistency and nonparametric methods for adversarial robustness.

Finally, the writing style is generally convoluted and requires substantial effort to parse. The abstract sentence starting with "In this work" and first sentence of the intro are both examples of this. The problem is too pervasive for me to give exhaustive pointers, but the paper requires substantial copyediting before it is accepted or published.

Minor / trivial:

Abstract - "In this work ..." has way too many commas.
Please use proper left quotes for Latex.

---

> ### Author Response · Authors · 2020-11-17
> **Official comment on the criticism of reviewer 3.**
>
> We want to thank reviewer 3 for the provided criticism and working out the positive and negative points, as well as the explanations for the given rating. We will comment on the "Negatives"-paragraphs in the given order:
>
> In terms of performance on the whole test set, one should indeed use the whole training set in these cases. However, the point of our work is that by aligning the robustness distributions of training and test data, one can improve model performance. Even by downsizing the training data, one can achieve a beneficial outcome. Hence, if one is interested in boosting accuracy on less robust test data (e.g. visually close images), it may be beneficial to remove the most robust training data.
>
> As other reviewers have pointed out as well, our work lacks a theoretic justification for why one would use the proposed concept in the first place. Furthermore, there were no definite explanations for the empirical data. Both of these we added in the updated version. We also want to thank the reviewer for the interesting references given. The notions in (Yang 2019) are indeed quite similar to those in our work. However, we focus on relations to general model performance and not adversarial robustness in particular.
>
>
> Again, we want to thank reviewer 3 for the provided criticism and remarks, and we hope that our mentioned corrections will motivate a second round of criticism.

---

### Official Review · AnonReviewer4 · 2020-10-28
**Official Blind Review #4**

**Rating:** 2
**Confidence:** 4

**Review:**

This work investigates how to choose the right training set and hyper-parameter for a test set. The authors claim that they introduce a concept of sample robustness based on the Lipschitz constant of the label map. The authors then empirically evaluate the robustness distribution of two datasets and investigate the model performance when using training subsets from different parts of the robustness distribution.

Pros:
- The idea of robustness distribution for a dataset seems interesting. It might be used to compare different datasets.
- The authors provide extensive experiments using different models. It is nice to see the details for each model and the robust samples for each dataset.

Cons:
- The novelty of this work is somewhat limited. The notion of sample robustness that the authors propose is just a minor modification of Lipschitz constant. Moreover, there is no theoretical or empirical justification regarding why this definition of sample robustness is useful.

- Despite that I really appreciate the efforts that the authors have put on the experimental section, I think that the contribution to the community is somewhat weak. This paper just uses the so called sample robustness to select a subset of the training sample and then reports the performance of model trained with that subset on some test data. But the authors do not provide any insights on how it works. In fact, from the results, it is still not clear how sample robustness can help choose the right training set and improve the model performance.

- There are some typos. For example, in the nine line of abstract, “be regarded as an intrinsic” should be “be regarded as being intrinsic”; in the related work, when citing multiple work, it is better use the command “\citep{work1, work2}”.

---

> ### Author Response · Authors · 2020-11-17
> **Official comment on the criticism of reviewer 4.**
>
> We want to thank reviewer 4 for the provided criticism working out the pros and cons, as well as the explanations for the given rating. We will comment on the cons in the given order:
>
> 1. Indeed, sample robustness is merely a normalised point-wise Lipschitz constant of the label map. However, this is precisely what we intended to create in the first place: an accessible measure of data-robustness that only depends on the data itself. The novelty of our work is not connecting Lipschitz calculus to robustness in general, but using this connection to analyse the robustness of datasets and its relation to model performance. Nonetheless, the lack of a theoretical motivation only allows for an empirical focus of our work and the reporting of experiments without proper explanations. Therefore, we have added two new sections in the updated version discussing our original motivation for this concept and natural relationships with KNN models.
>
> 2. As mentioned in 1. the proposed concept does indeed lack a theoretical justification. It is provided in the updated version.
>
> 3. We have reworked the entire text corpus to eliminate typos and allow for a better reading experience. (We meant: “be regarded as an intrinsic [...] feature”.)
>
> Again, we want to thank reviewer 4 for the provided criticism and remarks. We hope that our mentioned corrections will motivate a second round of criticism.

---

### Official Review · AnonReviewer1 · 2020-10-28
**Recommendation to reject**

**Rating:** 4
**Confidence:** 4

**Review:**

Summary:

This work introduces the concept of sample robustness – based on computing the pointwise Lipschitz constant of a data point – and use it to empirically analyze the effects of training on least and most robust training subsets on the performance for different models. This is done for both classification and regression setups. It is shown that the model performance can be sometimes improved by choosing particular training subsets and hyperparameters depending on the robustness distribution of the test (sub)sets.

-------------------------------------------------------------------
Pros:

-	The problem studied is interesting and is very relevant from a practical point of view.
-	The paper is structured well, and the problem has been motivated well in the introduction.
-	The related work and experiment results sections are quite thorough, and the empirical results are interesting.

------------------------------------------------------------------
Cons:

-	My main concern is that the definition of the sample robustness (in Section 3) seems overtly simplistic and ad-hoc. I understand the underlying reasoning behind the choice, but one could also consider many other definitions. It would have been more satisfactory had some supporting theoretical results been provided for the proposed notion.
-	The grammar and quality of writing can be improved at many places.
-	In Table 3, it seems that training on the complete training set gives the best loss/accuracy when tested on the complete test set (compared to when we train on subsets of the data). But then, what is the benefit here of training on a (least/most robust) subset of the training data? No explanation is provided in this regard.
-	In Table 4 as well, downsizing x drastically increases the loss on the complete test t. However, in Table 6 (KNN), choosing the 40,000 most robust training samples gives the best accuracy on t. There is no explanation provided as to why this is the case.

----------------------------------------------------------------------
Further remarks:

-	In Section 4.2: 50.000 --> 50,000 (same for 10.000).
-	It will be helpful for the reader if Algorithm 2 were in the main text.
-	In the experiments, it is not clear to me why the least and most robust test subsets are considered. In practice, we do not have access to the test data so we cannot process it.

--------------------------------------------------
Post rebuttal:
I have read the authors response and appreciate the effort made to improve the paper. But I think the results still need additional work, especially from the theoretical front. So my original score is unchanged.

---

> ### Author Response · Authors · 2020-11-17
> **Official comment on the criticism of reviewer 1.**
>
> We want to thank reviewer 1 for the criticism and the provided list of pros and cons. We are delighted to read that our work sparked interest in the proposed concept.
>
> As to the cons, we will comment in the given order:
>
> 1. As mentioned by other reviewers as well, our work lacks the theoretic motivation for the proposed concept of sample robustness. Furthermore, it seemed overly simplistic and may not convince a reader (there may be many equivalent choices). Indeed, sample robustness is merely a normalised point-wise Lipschitz constant of the label map. However, this is exactly what we intended to create in the first place, i.e. an accessible measure of data-robustness that only depends on the data itself. To give it a more theoretical basis, we have added a corresponding section in the updated version explaining our original motivation behind it. Furthermore, we have worked out the particular relationships of our concept to KNN models.
>
> 2. We did rewrite almost all paragraphs to enable a better reading flow.
>
> 3. The best performance on the whole test set ist achieved using the whole training set. The critical insight here, however, is that for the least- and most robust subsets of the test data, the model can achieve better performances (compared to the baseline) by using only 80% of the training set. It shows, for example, that such a model can generalise better on less robust data when trained on such data. One can therefore benefit from a collection of expert-models, each trained on a subset of different robustness.
>
> 4. Table. 4 indicates that there exists a model-dependent trade-off between (i) aligning robustness distributions by downsizing the training data and (ii) the overall ability of a model to generalise from the smaller set of samples. For KNN, we have provided additional theoretic relations and an explanation of precisely these results in the updated version. We attribute the outcome to the elimination of natural adversarial examples.
>
>
> Further remarks:
> - We used the correct writing now.
> - The algorithms are moved to the appendix as they build on additional theoretic results (also noted in the appendix).
> - We did use the most- and least robust subsets to amplify any noticeable effect.
>
> Again, we want to express our gratitude towards reviewer 1 for the provided criticism and remarks. Moreover, we hope that our mentioned corrections will motivate a second round of criticism.

---

### Official Review · AnonReviewer2 · 2020-10-29
**The works in this paper do not match with the announced contribution**

**Rating:** 5
**Confidence:** 3

**Review:**

This paper proposes a novel concept of sample robustness, which is used to find tailored training(sub)sets and hyperparameters depending on the robustness distribution of the test(sub)sets for boosting model performance. The motivation is straight forward and the proposed concept of sample robustness is inspiring. However, the works in this paper do not match with the announced contribution. Hence, I vote to reject.

Reasons for scores:
1，this paper announces that the novel concept of sample robustness can be used to choose training subsets and boost the model performance. In addition, some simple theoretical analysis about sample robustness are also given. But, this paper just gives some simple results of theoretical analysis and have no any explanation about those analytical results. More significantly, those analysis have little relationship with the goal of this paper. So this paper lacks some key theoretical analysis about sample robustness for its advantages.

2，The main works of this paper is based on empirical studies. Because of the lack of necessary theoretical analysis, more datasets are needed for verifying the behavior of model based on the concept of sample robustness. However, the datasets in this paper are too few to guarantee the good behavior of proposed technique in orther datasets.

3，this paper does not analyze the pros and cons of proposed technique based on the concept of sample robustness.

4，Some irregular use of mathematical symbols and grammatical errors.

Overall, the works in this paper do not support the announced contribution

---

> ### Author Response · Authors · 2020-11-17
> **Official comment on the criticism of reviewer 2.**
>
> We want to thank reviewer 2 for the provided criticism and the explanations for the rating. We will comment on them in the given order:
>
> 1. We have indeed given little theoretic motivation for our proposed concept and similarly few theoretic explanations of the empirical results. It prevents an exact match of what we proposed and what we delivered in this work. To this end, we have added additional theory in the updated version, which we furthermore used to explain the experimental data. Moreover, we changed the wording of our contribution to make it match with our present work.
>
> 2. As this work was intended to be an introduction, we only used the minimal setup of two different datasets to cover a classification- and regression setup. We intended to motivate readers to investigate the proposed concept in the future. However, as other reviews mentioned as well, the lack of additional theory caused our work to solely rely on this empirical data. Therefore, by adding such theory in the updated version, we hope to give a more convincing introduction.
>
> 3. Indeed, our work emphasised reporting of effects, rather than explaining them in the first place. In the updated version, we discuss the theoretic motivation of training on more- or less robust samples (in particular for KNN). Also, we mention a trade-off between the ability of a model to generalise and creating more similar robustness distributions by downsizing the training data.
>
> 4. We corrected the (mathematical) errors.
>
> Again, we want to thank reviewer 2 and hope that our mentioned corrections will motivate a second round of criticism.

---

### Author Response · Authors · 2020-11-17
**Official Comment and Changelog**

We first want to thank all reviewers for their thorough and constructive criticism as it already motivated many discussions and ideas for us to improve our work. After analysing the weak points mentioned and explained in the reviews, we made some non-trivial changes. A changelog can be found at the end.

The lack of theoretical justifications for the proposed concept and the missing explanations of the empirical results were among the main points of criticism. The reviewers rightfully criticised the present emphasis on experiments without a proper theory explaining these. More so, the few mathematical notions used to introduce sample robustness failed to motivate the purpose of the concept itself and seemed rather "ad hoc". To this end, we provided a thorough theoretical discussion of what motivated us to consider the robustness of samples initially. We also mentioned a trade-off between aligning robustness distributions and the ability of a model to generalise. Furthermore, we added a short paragraph discussing the relations between sample robustness and KNN models. With these additions to our current work, we then explained the empirical results and worked out similarities between the classification- and regression setup.

Another point of criticism was the general language style which decreased readability significantly. Here, we did a complete overhaul of all sections in order to enable a smoother reading flow.



Again, we want to thank all reviewers for their criticism and hope that they will find the motivation and time to post a second round of reviews helping us to improve our work further!



Changelog:

Removed:
- a table with less critical/interesting results for the ResNet-56
- parts of the additional theory in the appendix as it overall did not contribute to the work

Added:
+ a section to provide a theoretic discussion to motivate the concept
+ a section working out theoretic relations of sample robustness to KNN models, combined with an analysis of likely effects when using more- or less robust reference sets
+ theoretic explanations for the experimental results

Miscellaneous:
* Reworked almost all paragraphs to allow for a better reading experience, i.e. shortened sentences, decreased the number of commas, corrected typos and grammatical errors.
* Combined observations for the small CNN and the ResNet-56.

---

### Decision · Program_Chairs · 2021-01-07
**Final Decision**

**Decision:**

Reject

**Comment:**

The paper proposes sample robustness (a data-dependent measure) which is essentially a point-wise Lipschitz constant of the label map. The measure is used to choose a subset of training data for training and it measures how small of a perturbation is required to cause a label change w.r.t. label map.
initially, the paper lacked theoretical motivation and backing and the empirical studies were limited to be convincing enough. The authors added additional theoretical explanations. There were some mathematical mistakes that were fixed in the revision.
However, that is not enough to justify the proposal fully. Therefore, I suggest the authors improve the theoretical explanation. The paper would also benefit from more empirical analysis and discussion. As is, the paper has limited significance to the community since the conclusion is not convincing enough.


The paper writing quality although improved from the original version, still has room for improvement.

The proposed measure is simple, which can be a plus. but that means that we are also missing on some relationships and interactions between samples impact on training. Therefore, The paper will benefit from clearly discussing pros and cons of the proposed method. Moreover, discussing how this definition works in choosing the best subset of samples will improve the paper.

i thank the authors for their effort and improving the paper in response to the reviews. However, given that myself and reviewers find the modifications enough for addressing all the concerns, I vote for rejecting the paper. Please improve the paper and resubmit to a future venue.